# How Can LLMs Serve as Experts in Malicious Code Detection? A Graph Representation Learning Based Approach

## Abstract

Large Language Models (LLMs) excel in code processing yet encounter challenges in malicious code detection, primarily due to their limited ability to capture long-range dependencies within large and complex codebases. To address this limitation, we propose a graph representation learning-based attention acquisition framework to enhance LLMs' malicious code detection capabilities. Specifically, our method constructs a graph representation of the code, extracts semantic and structural features using an LLM, and trains a Graph Neural Network (GNN) with minimally labeled data. The GNN performs an initial detection and, through backtracking of its predictions, identifies key code sections that are most likely to contain malicious behavior. These identified sections then guide the attention of the LLM for in-depth analysis. By concentrating the LLM's processing on these critical regions, our approach reduces the interference of redundant or irrelevant data, thereby improving detection accuracy and efficiency while maintaining low annotation costs. Extensive evaluation on both custom-built and public datasets demonstrates that our approach consistently outperforms existing detection methods, highlighting its potential for practical deployment in software security scenarios. *Code and data can be found in the Supplementary Materials.*

## 1 Introduction

LLMs excel at handling code-related tasks (Nam et al., 2024; Haroon et al., 2025), including code understanding (Li & Shin, 2024), generation (Fan et al., 2023), and logic optimization (Ishibashi & Nishimura, 2024). Current LLMs can generate efficient and highly readable code based on requirements, thanks to their deep understanding of a wide range of programming languages and syntax. Through natural language processing techniques, LLMs can also effectively identify syntax errors and potential vulnerabilities in code snippets and provide developers with suggestions for fixes, thereby improving code quality (Huynh et al., 2025). However, in the field of code security—particularly in malicious code detection—LLMs face limitations, as they are not yet capable of analyzing entire code projects with the same depth and comprehensiveness as security experts or professional tools, especially when it comes to large-scale projects (Wang et al., 2025; Akuthota et al., 2023; Xue et al., 2024).

The above argument can be practically validated. Specifically, we conduct tests on some collected Python packages. The experimental results are shown in Figure 1. As seen, the overall analysis performances of the LLM baselines are still far from those of professional analysis tools, especially for large packages. Another obvious observation is that, as the size of the package increases, the performance of LLM in detection decreases. We believe this is primarily due to the inherent characteristics of LLMs. When dealing with large amounts of code, LLMs' attention mechanisms often struggle to accurately pinpoint problematic snippets, instead wasting attention on large portions of benign code (Hu et al., 2025). This explains why, in the experimental results shown in the figure, the larger the program, the worse the performance of the LLM analysis. Additionally, when the internal relationships within the code are complex and involve multiple interdependencies, LLMs face even greater difficulty. They struggle to identify malicious behaviors that are present in call relationships, especially when the relevant code snippets are not sequentially close. This also hinders their ability to properly attend to and jointly analyze these distant, yet connected, pieces of code.

Furthermore, when LLMs are used to analyze a large number of code projects, they typically require significantly more computational resources compared to conventional methods. Unlike typical tools that operate based on specific rules or smaller, specialized neural networks, LLMs rely on their large-scale models and the vast amount of knowledge they contain to identify malicious code. This means that for each code project, especially at a large scale, resource consumption—such as memory and processing time—can increase rapidly. As a result, LLMs are less efficient in scenarios that require large-scale code analysis.

We aim to address these issues and enhance the capabilities of LLMs in malicious code detection, striving to develop a solution that surpasses professional tools in performance while being cost-effective, thereby highlighting the practical value of our research. To this end, our research focuses on Python. As an interpreted language, its nature, which is closer to that of a natural language, is inherently compatible with the sequence-processing mechanisms of LLMs (Wang et al., 2021; Nijkamp et al., 2023), making it an ideal subject for validating our approach. Additionally, Python code and technical documentation constitute some of the most abundant sources of code training data for LLMs, which leads to enhanced analysis capabilities (Chen et al., 2021; Rozière et al., 2023). Therefore, towards Python code, we propose a ***Graph Representation Learning-Enhanced Malicious code detection framework for LLM***, abbreviated as GMLLM. GMLLM utilizes a graph representation learning approach to identify critical sections within projects that require closer inspection, which serves as attentions that allow the LLM to focus on these key areas, reducing redundant information and improving detection performance.

GMLLM first constructs graph representations of the input Python packages, using an LLM to extract feature vectors for each node. It then trains a GNN model with minimally labeled data that indicates whether a package contains malicious code. This trained model provides an initial, albeit potentially inaccurate, assessment of new packages. We then introduce an explanation paradigm to identify

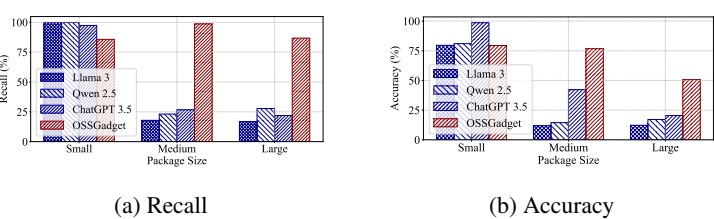

(a) Recall          (b) Accuracy

Figure 1: Comparison between LLMs directly applied to malicious code detection and a commonly used malicious code detection tool OSSGadget (Microsoft, 2019). The blue bars represent the performance of the LLMs, while the red bar represents the performance of the tool. Large, Medium, Small refer to the scale of packages.

and measure the influence of input graph nodes on the GNN's classification. If a node has a significant influence on the GNN's decision to classify a sample as malicious, this suggests that the node itself possesses features characteristic of malicious code. We then filter these influential nodes and use an LLM for a more detailed analysis. This approach both fully leverages the LLM's code understanding capabilities and avoids the performance issues associated with analyzing entire packages. Furthermore, the method requires only binary labels (i.e., malicious or not) for training, which allows it to make full use of abundant relevant data. We compare this approach with multiple LLM baselines and other malicious code detection methods and demonstrate its superiority.

Our contributions are as follows:

- We propose a novel paradigm to address the challenges LLMs face in detecting malicious code, thereby fostering the broader application of LLMs in software security-related domains.

- Focusing on Python programs, we design the GMLLM framework to implement the proposed paradigm, which significantly enhances the malicious code detection capabilities of LLMs.

- We validate our approach using a proposed comprehensive Python malicious code dataset and multiple public datasets, demonstrating the effectiveness of our method.

## 2 RELATED WORKS

### 2.1 MALICIOUS CODE DETECTION

Malicious code detection is vital for information security, with conventional methods including metadata-based, rule-based, and learning-based approaches. Metadata-based methods, such as package name analysis (Kaplan & Qian, 2021), are easily bypassed. Rule-based methods, like Yara (VirusTotal, 2023) and Bandit (PyCQA, 2023), are more precise but require expert-designed rules, making them resource-intensive (Duan et al., 2021). Learning-based methods using deep learning have achieved high accuracy, extracting features such as API calls and opcodes (Yadav et al., 2022; Aamir et al., 2024; Wu et al., 2023), but face challenges with adversarial attacks (Yumlembam et al., 2023; Amin et al., 2022). In the meantime, within open-source ecosystems like PyPI, malicious code attacks still cause severe disruptions and losses (Okafor et al., 2022). While machine learning models have achieved near-perfect detection performance (Dambra et al., 2023; Liang et al., 2023; Sun et al., 2024), challenges remain regarding methodology, dataset design, and false positive tolerance (Arp et al., 2022; Vu et al., 2023).

### 2.2 LLM FOR MALICIOUS CODE DETECTION

There is a growing trend toward leveraging LLMs for the detection of malicious code (Zahan et al., 2025). Fang et al. (2024) find that LLMs perform well on simple code but struggle with complex or obfuscated code. Yu et al. (2024) enhances detection accuracy by combining LLMs with static analysis. Akinsowon & Jiang (2024) show LLMs improve detection rates for malicious code challenging for static analysis. LLMs also address malicious code sample scarcity by generating diverse datasets (Yu et al., 2024), but their generative capabilities pose security risks. Khan et al. (2024) propose an integrated framework combining LLMs with traditional techniques to effectively address malicious code detection. Despite the advances, LLMs still face difficulties with large-scale code projects. Our work aims to address this issue.

## 3 METHOD

The framework of our proposed GMLLM is shown in Figure 2 and can be divided into two parts. Part one involves training a GNN using a dataset of malicious code with simple annotations. Part two interprets the trained GNN on new samples to obtain attention, which then guides the LLM to perform malicious code detection. Below, we will introduce these two parts separately.

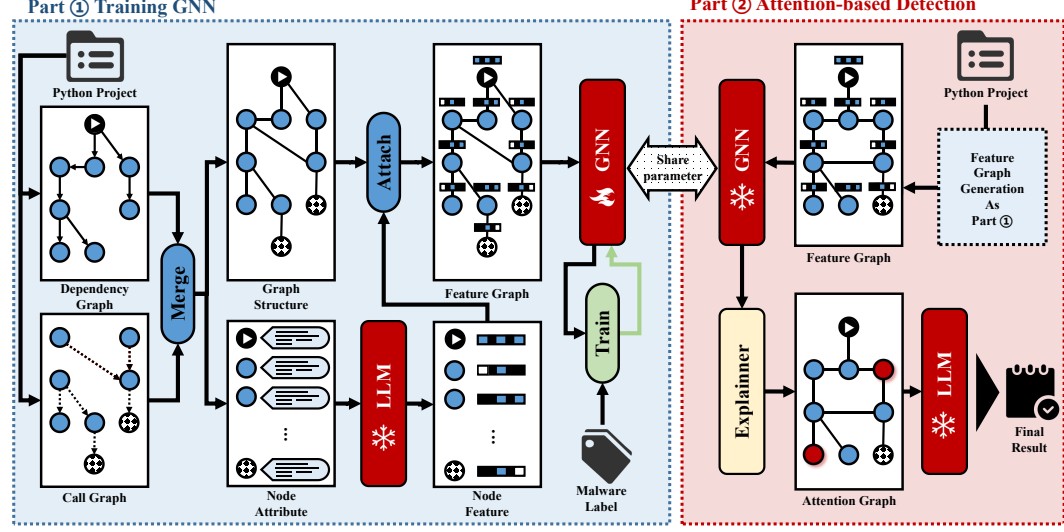

Figure 2: Framework of the proposed GMLLM.

## 3.1 Training GNN

To facilitate subsequent processing using GNNs, we represent the source code of each Python project as a graph $G^{\text{code}} = \{\mathcal{V}^{\text{code}}, \mathcal{E}^{\text{code}}\}$, where $\mathcal{V}^{\text{code}}$ is the set of nodes and $\mathcal{E}^{\text{code}}$ is the set of edges.

The node set $\mathcal{V}^{\text{code}}$ is derived from the Abstract Syntax Tree (AST) of the Python project. We traverse all the ".py" files and parse the source code into AST objects. The nodes in $\mathcal{V}^{\text{code}}$ correspond to key entities in the Python code, such as classes, functions, and modules. Each node is annotated with the source code corresponding to the respective element. The edge set $\mathcal{E}^{\text{code}}$ includes dependency edges and call relationship edges. Please refer to **Appendix C.1** for detailed implementations.

Next, we extract the features of each node $v$ in $\mathcal{V}^{\text{code}}$ and represent them as a vector $h_v$. The feature extraction process involves defining a set of sensitive behavior rules $\mathcal{S}$, which are used to capture the sensitive behavior features of the code corresponding to each node $v$. Such a rule-matching approach is commonly used in Python malicious code detection. What sets this approach apart is that we have employed LLM for automated rule generation and design the corresponding rule form.

The set $\mathcal{S}$ can be divided into two parts: common sensitive behavior rules $\mathcal{S}^{\text{comm}}$, and data-derived sensitive behavior rules $\mathcal{S}^{\text{data}}$. $\mathcal{S}^{\text{comm}}$ is generated as follows:

$$\mathcal{S}^{\text{comm}} = \text{LLM}(\rho^{\text{comm}}),$$

where $\text{LLM}(\cdot)$ represents the LLM model, and $\rho^{\text{comm}}$ is the prompt used. $\rho^{\text{comm}}$ instructs the LLM to summarize common sensitive behavior rules and output them as a list.

The rules in $\mathcal{S}^{\text{data}}$ are obtained by sampling 10% of the training data and passing the corresponding code through the LLM to generate the data-related sensitive behavior rules:

$$\mathcal{S}^{\text{data}} = \bigcup_{i=1}^{m} \text{LLM}(\rho^{\text{data}}, X_i)$$

where $\{X_i\}_{i=1}^{m}$ represents the set of Python files collected from the training data. Details concerning the implementations and prompts are offered in **Appendix C.2**.

After obtaining $\mathcal{S}$, a feature vector $h_v$ of dimension $|\mathcal{S}|$ is constructed for each node $v$. This vector is a multi-hot vector where each position indicates whether a rule in $\mathcal{S}$ is matched. This method automates the extraction of node-sensitive behavior features and is highly scalable. When new scenarios arise, the system only requires the addition of new environmental samples to automatically update the sensitive feature rules.

Based on the collected graph and node features, we proceed to train a GNN $g_\theta(\cdot)$. As discussed in the introduction, we formulate the task as a binary classification problem in order to minimize the need for labeled data. Following the aforementioned transformations, we obtain $n$ graph samples, denoted as $\{G_i^{\text{code}}\}_{i=1}^{n}$, where $n$ is the number of Python projects in the training set. These samples are then processed using a GNN. In our implementation, we employ a two-layer Graph Convolutional Network (GCN), though it is worth noting that any GNN model capable of handling graph-structured data could be applied to this task.

The training process follows a standard binary classification framework, with cross-entropy loss used for optimization. Formally, it can be expressed as:

$$\mathcal{L}(\theta) = -\frac{1}{n} \sum_{i=1}^{n} \left[ y_i \log \left( g_\theta(G_i^{\text{code}}) \right) + (1 - y_i) \log \left( 1 - g_\theta(G_i^{\text{code}}) \right) \right] \tag{1}$$

## 3.2 Attention Based Detection

We have developed a GNN model designed to detect potential malicious code based on sensitive behavioral features. This model is used to classify the target package, after which we conduct interpretability analysis on the classification results to identify the key nodes that lead the GNN to classify a package as malicious. In real-world projects, malicious code segments typically constitute only a small fraction of the overall codebase. Therefore, this approach can effectively pinpoint suspicious nodes, enabling an LLM to perform targeted analysis on these critical areas. In essence, we locate which specific nodes "led" the GNN to classify the target package as malicious, then focus

on analyzing those nodes to verify whether they indeed represent malicious code and to understand their malicious behavior. The key advantage of this approach is that it provides the LLM with a focused attention mechanism for the entire project at relatively low training and inference costs, while avoiding the accuracy and resource consumption issues associated with having the LLM directly process the whole package.

Now we give a detailed presentation. For the case where the $j$-th sample under test, $G_j^{\text{code}}$, is identified as containing malicious code, we will specifically construct trainable edge mask $M_j^{\text{edge}}$ and a feature mask $M_j^{\text{feat}}$ on the target graph $G_j^{\text{code}}$. Let the adjacency matrix and feature matrix of $G_j^{\text{code}}$ be $A_j \in \{0,1\}^{|\mathcal{V}_j| \times |\mathcal{V}_j|}$ and $H_j \in \mathbb{R}^{|\mathcal{V}_j| \times |\mathcal{S}|}$. For the target graph $G_j^{\text{code}}$, we then set $M_j^{\text{edge}} \in \mathbb{R}^{|\mathcal{V}_j| \times |\mathcal{V}_j|}$ and $M_j^{\text{feat}} \in \mathbb{R}^{|\mathcal{V}_j|}$. Apply a sigmoid activation $\delta(\cdot)$ to $M_j^{\text{edge}}$ and symmetrize:

$$\dot{M}_j^{\text{edge}} = \delta(M_j^{\text{edge}}), \qquad \tilde{M}_j^{\text{edge}} = \frac{\dot{M}_j^{\text{edge}} + (M_j^{\text{edge}})^\top}{2}, \tag{2}$$

then remove the diagonal and perform an element-wise product with the original graph to obtain the *masked adjacency*:

$$\hat{A}_j = \left(A_j \odot \tilde{M}_j^{\text{edge}}\right) \odot (\mathbf{1} - I), \tag{3}$$

where $\odot$ denotes the Hadamard product. On the feature side, use $\tilde{M}^{\text{feat}} = \sigma(M^{\text{feat}})$ and broadcast:

$$H_j' = \text{diag}(\tilde{M}_j^{\text{feat}})H_j. \tag{4}$$

Here, $\text{diag}(\tilde{M}_j^{\text{feat}})$ denotes a diagonal matrix whose diagonal entries are the elements of $\tilde{M}_j^{\text{feat}}$. Next, we feed $(H_j', \hat{A}_j)$ into the trained GNN $g_\theta(\cdot)$ to obtain the predicted probabilities $p_j$ for sample $G_j^{\text{code}}$. $p_{j,(0)}$ and $p_{j,(1)}$ denotes the predicted probability of $g_\theta(\cdot)$ on $G_j^{\text{code}}$ as benign and malicious respectively. Then, we update the trainable masks using the following loss:

$$\mathcal{L}_{\text{pred}} = \log\left(1 - p_{j,(1)}\right), \tag{5}$$

where $\mathcal{L}_{\text{pred}}$ encourages the mask to only preserve the substructure that could lead the $G_j^{\text{code}}$ to be classified as malicious. To encourage sparsity and determinacy, we add size regularizers on the edge and feature masks:

$$\mathcal{L}_{\text{size}} = \|\tilde{M}_j^{\text{edge}}\|_1 + \|\tilde{M}_j^{\text{feat}}\|_1, \tag{6}$$

and entropy regularizer:

$$\mathcal{L}_{\text{ent}} = \frac{1}{|\tilde{M}_j^{\text{edge}}|} \sum_{m_{(k,l)} \in \tilde{M}_j^{\text{edge}}} \left( - m_{(k,l)} \log m_{(k,l)} - (1 - m_{(k,l)}) \log(1 - m_{(k,l)}) \right). \tag{7}$$

The overall objective is:

$$\mathcal{L} = \mathcal{L}_{\text{pred}} + \lambda^{\text{size}} \mathcal{L}_{\text{size}} + \lambda^{\text{ent}} \mathcal{L}_{\text{ent}}. \tag{8}$$

$\mathcal{L}$ is optimized through backpropagation, yielding $M_j^{\text{edge}}$ and $M_j^{\text{feat}}$ as the explanatory mask. $\lambda^{\text{size}}$ and $\lambda^{\text{ent}}$ are hyperparameters. Note that $\mathcal{L}$ is computed and optimized individually for each sample, with the objective of finding the appropriate mask. The masks $\tilde{M}_j^{\text{edge}}$ and $\tilde{M}_j^{\text{feat}}$ effectively reflect the graph structures that lead the GNN to classify the sample as containing malicious code, as these nodes maximize $p_{j,(1)}$, the predicted probability of $g_\theta(\cdot)$ on $G_j^{\text{code}}$ being malicious. This is precisely the part that should be analyzed in detail using LLMs. Therefore, $\mathcal{L}$ is applied on a per-sample basis.

We set the value of the attention score for each element of $G_j^{\text{code}}$ as the corresponding mask value within $\tilde{M}_j^{\text{edge}}$. Formally, we have the node attention score $a_v$ as:

$$a_v = (\tilde{M}_j^{\text{feat}})_v, \tag{9}$$

and edge attention score $a_{v,w}$ as:

$$a_{v,w} = (\tilde{M}_j^{\text{edge}})_{v,w}. \tag{10}$$

Based on the acquired attention score, we employ an LLM to perform more targeted and in-depth analysis. We set a threshold $\alpha$, and only the code graph structures with scores greater than $\alpha$ are fed into the LLM for malicious code detection. In practice, since the majority of the code is benign, this step can filter out a large amount of irrelevant code, significantly saving computational resources and enabling the LLM to process the problem more efficiently. We denote the extracted subgraph as $\mathrm{Att}(\cdot)$, which needs to be processed by the LLM. $\mathrm{Att}(\cdot)$ can be formulated as:

$$\mathrm{Att}(G_j^{\mathrm{code}}) = \Big( \big\{ v \in \mathcal{V}_j^{\mathrm{code}} | a_v < \gamma^{\mathrm{node}} \big\}, \big\{ (v, w) \in \mathcal{E}_j^{\mathrm{code}} | a_{v,w} < \gamma^{\mathrm{edge}} \big\} \Big). \tag{11}$$

In the implementation, in order to convert the graph-structured data into a natural language description that can be understood and processed by the LLM, we combine the information from both the AST and the source code. Based on the structure of $\mathrm{Att}(G_j^{\mathrm{code}})$, we provide segmented descriptions of the nodes, edges, and the content of the nodes. A concrete example is given below:

---

**Examples of Att($G_j^{\mathbf{code}}$)**

**Nodes:**
15Cent-999.0.1.setup
15Cent-999.0.1.setup.CustomInstal1
......
**Edges:**
10Cent11-999.0.4.setup.CustomInstall → 10Cent11-999.0.4.setup.CustomInstall.run
10Cent11-999.0.4.setup → 10Cent11-999.0.4.setup.CustomInstall
......
**Node Attributes:**
Codes of 15Cent-999.0.1.setup are
  " import re, sys, pathlib
   indicators = r"""
......

---

Based on $\mathrm{Att}(G_j^{\mathrm{code}})$, the overall process can be formally expressed as follows:

$$(\hat{Y}, T) = \mathrm{LLM}\Big( \rho^{\mathrm{ana}}, \mathrm{Att}(G_j^{\mathrm{code}}) \Big), \tag{12}$$

where $\hat{Y}$ denotes the predicted classification, $T$ denotes the detailed description of location and characteristics concerning the malicious codes and will be left blank if $G_j^{\mathrm{code}}$ is classified as benign, and $\rho^{\mathrm{ana}}$ denotes the utilized prompt. Details concerning $\rho^{\mathrm{ana}}$ are given in **Appendix** C.3.

Next, our method can output the final results using the obtained $\hat{Y}$ and $T$ from equation 12. Compared to other commonly used tools and methods, our approach not only detects the presence of malicious code but also provides specific description of its characteristics and locations.

## 4 EXPERIMENTS

In this section, we conduct multiple experiments with our proposed GMLLM and other baselines to answer the following research questions (RQs):

**RQ1:** Does GMLLM lead to a significant improvement in the detection of malicious code by LLMs?

**RQ2:** Is the performance of GMLLM superior to that of conventional malicious code detection tools?

**RQ3:** Can GMLLM provide a comprehensive analysis of malicious code?

**RQ4:** Does GMLLM help in reducing the computational overhead of LLMs for malicious code detection?

### 4.1 SETTINGS

We performed a series of experiments using multiple datasets to validate the effectiveness of our method. Specifically, we utilized three publicly available datasets: Backstabbers (Ohm et al., 2020), Datadog (Guarddog, 2023), and Mal-OSS (Guo et al., 2023). Additionally, we created a larger-scale dataset, the Malicious Codes from PYPI (MalCP) dataset, by incorporating samples collected from PYPI. The MalCP dataset is partitioned into three subsets based on program size: Large, Medium, and Small. An additional All category is included to represent the entire set of program samples. The MalCP dataset also incorporates detailed descriptions of malicious behaviors into its benchmark to evaluate the performance of LLM-based methods. In

addition, MalCP covers a diverse set of attack tactics, with three major categories (Execution, Impact, Credential Access/Collection) and a long tail of others, so that no single tactic dominates the dataset. Detailed information about the MalCP dataset can be found in **Appendix D.1**.

For the comparison with the baselines, we evaluated the performance of our method against various LLMs and malicious code detection methods and tools. Specifically, we selected multiple different LLMs for direct application in malicious code detection, including Qwen 2.5 (Yang et al., 2024), Llama 2 (Touvron et al., 2023), Llama 3 (Dubey et al., 2024), ChatGPT 3.5 (Ye et al., 2023), and ChatGPT 4o (Hurst et al., 2024). Additionally, we chose three rule-based common malicious code detection tools: Bandit4Mal (Vu, 2020), OSSGadget (Microsoft, 2019), and Virustotal (Hispasec Sistemas, 2004), as well as two language model- and neural network-based malicious code detection tools: MPHunter (Liang et al., 2023) and Ea4mp (Sun et al., 2024). To ensure statistical robustness, all experimental results are reported as the average across five independent trials. We implemented our method using two base LLMs, Llama 3 and ChatGPT 4o, and compared their performance with the baselines. Detail settings can be found in **Appendix D.2**.

Table 1: Performance on Backstabbers, Datadog, Mal-OSS Datasets. **Bold** indicates the best performance, while underline indicates the second-best performance.

| Model | Backstabbers | | | Datadog | | | Mal-OSS | | |
|---|---|---|---|---|---|---|---|---|---|
| | Recall | Precision | ACC | Recall | Precision | ACC | Recall | Precision | ACC |
| ChatGPT 3.5 | 28.45 | 37.19 | 44.66 | 54.30 | 48.32 | 56.30 | 73.34 | 60.14 | 65.29 |
| ChatGPT 4o | 92.89 | 90.65 | 92.28 | 80.07 | 84.73 | 85.53 | 89.69 | 90.22 | 90.76 |
| Llama 2 | 3.48 | 3.20 | 6.65 | 15.12 | 11.03 | 12.88 | 11.78 | 10.14 | 11.14 |
| Llama 3 | 24.96 | 22.45 | 25.39 | 43.99 | 30.33 | 33.86 | 72.31 | 45.17 | 46.74 |
| Qwen 2.5 | 28.30 | 25.10 | 27.74 | 49.83 | 33.49 | 37.19 | 78.50 | 47.46 | 50.00 |
| Bandit4Mal | 12.92 | 29.67 | 45.53 | 62.89 | 66.06 | 70.77 | 23.56 | 45.33 | 51.63 |
| OSSGadget | **96.81** | 64.51 | 73.88 | 86.60 | 57.40 | 67.29 | 86.01 | 59.47 | 66.51 |
| Virustotal | 87.52 | 84.81 | 86.97 | 82.13 | 82.70 | 85.24 | 46.69 | 76.39 | 68.75 |
| MPHunter | 93.12 | 89.81 | 91.08 | 93.38 | 90.17 | 94.67 | 86.14 | 83.06 | 88.50 |
| Ea4mp | 94.58 | 91.13 | 93.17 | 82.13 | 82.70 | 85.24 | 85.19 | 83.72 | 84.99 |
| GMLLM (Llama 3 based) | 96.52 | 74.30 | 82.94 | **97.94** | 73.83 | 84.52 | **98.82** | 68.89 | 78.87 |
| GMLLM (ChatGPT 4o based) | 93.03 | **94.54** | **94.29** | 97.25 | **95.61** | **96.96** | 96.76 | **95.36** | **96.33** |

Table 2: Performance on our proposed MalCP dataset. **Bold** indicates the best performance, while underline indicates the second-best performance.

| Model | Large | | | Medium | | | Small | | | All | | | Benign |
|---|---|---|---|---|---|---|---|---|---|---|---|---|---|
| | recall | precision | acc | recall | precision | acc | recall | precision | acc | recall | precision | acc | recall |
| ChatGPT 3.5 | 21.91 | 15.86 | 20.43 | 26.70 | 34.97 | 42.25 | 97.38 | **99.66** | 98.59 | 51.36 | 50.65 | 55.18 | 58.35 |
| ChatGPT 4o | 67.25 | 76.17 | 77.98 | 98.30 | 93.53 | 96.01 | 97.38 | 94.14 | 95.84 | 89.33 | 89.49 | 90.39 | 91.27 |
| Llama 2 | 7.59 | 5.22 | 5.68 | 2.55 | 2.27 | 2.56 | 16.07 | 16.23 | 20.09 | 8.92 | 7.64 | 9.64 | 10.24 |
| Llama 3 | 16.70 | 11.31 | 12.26 | 17.86 | 14.46 | 11.82 | **99.84** | 70.16 | 79.59 | 47.68 | 34.77 | 35.60 | 25.54 |
| Qwen 2.5 | 27.77 | 17.58 | 17.14 | 23.13 | 18.01 | 14.46 | **99.84** | 71.65 | 81.00 | 52.62 | 37.42 | 38.50 | 26.74 |
| Bandit4Mal | 64.21 | 51.48 | 60.57 | 9.35 | 27.64 | 45.93 | 13.28 | 51.92 | 52.59 | 26.04 | 46.45 | 52.77 | 75.01 |
| OSSGadget | 86.77 | 44.74 | 50.71 | **98.64** | 67.21 | 76.76 | 85.74 | 74.93 | 79.43 | 90.60 | 61.22 | 69.66 | 52.23 |
| Virustotal | 83.51 | 85.37 | 87.39 | 87.93 | 84.89 | 86.98 | 42.13 | 71.99 | 64.44 | 69.86 | 81.79 | 79.24 | 87.05 |
| MPHunter | 85.60 | 81.27 | 83.83 | 93.32 | 89.18 | 90.52 | 96.39 | 95.03 | 96.50 | 94.13 | 90.81 | 93.56 | 91.57 |
| Ea4mp | 87.31 | 86.23 | 88.10 | 91.76 | 92.63 | 92.79 | 97.19 | 96.74 | 97.00 | 93.12 | 92.17 | 91.03 | 92.37 |
| GMLLM (Llama 3 based) | **94.36** | 65.32 | 77.18 | 98.30 | 73.44 | 82.51 | 99.67 | 75.81 | 84.62 | **97.71** | 71.88 | 81.60 | 68.19 |
| GMLLM (ChatGPT 4o based) | 87.20 | **89.14** | **90.41** | 97.45 | **96.14** | **96.96** | 99.34 | 98.38 | **98.90** | 95.30 | **95.07** | **95.62** | **95.89** |

## 4.2 Malicious Code Detection Capability Comparisons (RQ1, RQ2)

We first compared the accuracy of malicious code detection. Tables 1 and 2 present the detailed results. As observed, our method demonstrates superior performance on both public datasets and our own constructed dataset, outperforming baseline methods in the majority of scenarios and metrics. Notably, the improvement of our method over LLMs is significant, especially in the Precision metric, highlighting the ability of GMLLM to filter out redundant benign code interference, thereby validating the effectiveness of GMLLM. Furthermore, the comparison with other tools also indicates that, based on GMLLM, the LLM-based approach can effectively handle malicious code detection, particularly in the domain of Python code.

Table 3: Model performances on describing the details concerning the malicious behaviors. **Bold** indicates the best performance, while underline indicates the second-best performance. Baselines including Bandit4Mal, OSSGadget, Virustotal, MPHunter, and Ea4mp are excluded as they are incapable of such description.

| Method | Threat Generality (↑) | Execution Path Traceability (↑) | Evidence Groundedness (↑) | Average Quality Score (↑) | Quality Score Standard Deviation (↓) | Factual Alignment (↑) |
|---|---|---|---|---|---|---|
| All | | | | | | |
| ChatGPT 3.5 | 2.013 | 1.500 | 1.681 | 1.731 | 1.699 | 1.523 |
| ChatGPT 4o | 3.427 | **2.764** | 2.967 | 3.053 | 1.166 | 2.602 |
| Llama 2 | 0.244 | 0.177 | 0.207 | 0.209 | 0.714 | 0.149 |
| Llama 3 | 1.731 | 1.256 | 1.564 | 1.517 | 1.605 | 1.325 |
| Qwen 2.5 | 1.995 | 1.120 | 1.510 | 1.541 | 1.487 | 1.420 |
| GMLLM (Llama 3 based) | 3.037 | 2.056 | 2.464 | 2.519 | **0.647** | 2.060 |
| GMLLM (ChatGPT 4o based) | **3.687** | 2.727 | **3.215** | **3.210** | 0.860 | **2.721** |
| Large | | | | | | |
| ChatGPT 3.5 | 0.842 | 0.633 | 0.705 | 0.727 | 1.385 | 0.640 |
| ChatGPT 4o | 2.716 | 2.167 | 2.336 | 2.406 | 1.732 | 1.978 |
| Llama 2 | 0.156 | 0.119 | 0.137 | 0.137 | **0.524** | 0.111 |
| Llama 3 | 0.584 | 0.432 | 0.527 | 0.514 | 1.165 | 0.453 |
| Qwen 2.5 | 1.017 | 0.640 | 0.777 | 0.811 | 1.349 | 0.714 |
| GMLLM (Llama 3 based) | 2.857 | 1.928 | 2.369 | 2.385 | 0.824 | 1.915 |
| GMLLM (ChatGPT 4o based) | **3.377** | **2.520** | **3.045** | **2.981** | 1.262 | **2.512** |
| Medium | | | | | | |
| ChatGPT 3.5 | 1.024 | 0.748 | 0.845 | 0.872 | 1.462 | 0.777 |
| ChatGPT 4o | 3.682 | **2.794** | 2.889 | 3.122 | 0.637 | 2.643 |
| Llama 2 | 0.073 | 0.051 | 0.063 | 0.062 | **0.397** | 0.039 |
| Llama 3 | 0.713 | 0.527 | 0.641 | 0.627 | 1.289 | 0.536 |
| Qwen 2.5 | 0.859 | 0.505 | 0.658 | 0.674 | 1.250 | 0.582 |
| GMLLM (Llama 3 based) | 2.917 | 1.959 | 2.287 | 2.388 | 0.585 | 1.879 |
| GMLLM (ChatGPT 4o based) | **3.755** | 2.770 | **3.155** | **3.226** | 0.672 | **2.661** |
| Small | | | | | | |
| ChatGPT 3.5 | 3.851 | 2.879 | 3.225 | 3.318 | 0.597 | 2.908 |
| ChatGPT 4o | 3.852 | **3.310** | **3.628** | **3.577** | 0.733 | **3.034** |
| Llama 2 | 0.474 | 0.343 | 0.398 | 0.405 | 0.984 | 0.285 |
| Llama 3 | 3.579 | 2.580 | 3.238 | 3.132 | 0.493 | 2.746 |
| Qwen 2.5 | 3.828 | 2.075 | 2.885 | 2.930 | 0.688 | 2.762 |
| GMLLM (Llama 3 based) | 3.289 | 2.246 | 2.707 | 2.747 | **0.463** | 2.283 |
| GMLLM (ChatGPT 4o based) | **3.853** | 2.842 | 3.399 | 3.365 | 0.562 | 2.935 |

Table 4: A comparison of the token counts used by LLM baselines in the MalCP dataset, categorized by program sample size (Large, Medium, and Small). The best performance is indicated in **bold**, while the second-best is underlined.

| | Large | | | | Medium | | | | Small | | | |
|---|---|---|---|---|---|---|---|---|---|---|---|---|
| | Mean | Std | Min | Max | Mean | Std | Min | Max | Mean | Std | Min | Max |
| Llama3 | 259089.13 | 907541.88 | 365 | 12810537 | 28578.12 | 16497.45 | 296 | 39274 | 698.47 | 188.77 | 257 | 2523 |
| ChatGPT 4o | 251001.56 | 868507.92 | 370 | 12875186 | 28866.54 | 17267.57 | 307 | 150823 | 700.48 | 187.67 | 269 | 2544 |
| ChatGPT 3.5 | 255755.00 | 882059.26 | 376 | 12813070 | 29466.69 | 17614.16 | 307 | 150331 | 713.57 | 189.33 | 269 | 2530 |
| Qwen 2.5 | 280463.08 | 972292.04 | 375 | 14226559 | 29371.04 | 16789.81 | 297 | 40199 | 706.27 | 197.05 | 261 | 2533 |
| Llama 2 | 333972.02 | 1207832.80 | 482 | 1954134 | 31678.23 | 17814.37 | 385 | 43165 | 852.12 | 249.55 | 332 | 3297 |
| GMLLM (Llama 3 based) | 644.30 | **408.97** | 201 | 2587 | 455.53 | 200.56 | **199** | 1661 | 299.80 | 254.70 | **204** | 1698 |
| GMLLM (ChatGPT 4o based) | **640.06** | 409.35 | 198 | 2418 | 451.31 | 199.03 | 203 | 1557 | 292.65 | 240.82 | 209 | 1625 |

### 4.3 MALICIOUS BEHAVIOR DETAILED DESCRIPTION COMPARISONS (RQ3)

To analyze the performance of our algorithm, we compared the model-generated descriptions of malicious behavior against the ground-truth descriptions in the MalCP dataset. This comparison was limited to LLM-based methods, as other approaches are incapable of generating such detailed descriptions. For metrics, we used the following metrics: 1) Threat Generality, which measures the ability to generalize from specific functions or code elements to recognized cybersecurity tactics; 2) Execution Path Traceability, which assesses the clarity of the reconstructed execution flow; 3) Evidence Groundedness, which examines how closely a conclusion is supported by specific code elements; 4) Average Quality Score, the average score of the first three metrics; 5) Quality Score Standard Deviation, the standard deviation of the average evaluation quality; and 6) Factual Alignment, which evaluates the accuracy of identified and described malicious behaviors against a ground truth. Threat Generality, Execution Path Traceability, Evidence Correlation, and Factual Consistency were evaluated by an LLM, with the details discussed in **Appendix C.4**.

The results, as shown in Table 3, indicate that our method achieves superior performance in the vast majority of scenarios. This is particularly evident on large and medium-scale code, where foundational LLMs, when used alone, are more susceptible to interference. To validate the reliability of using GPT-4o as an automatic evaluator for the explanation-quality metrics, we additionally performed a human evaluation study. Two security practitioners independently scored a subset of model explanations along the same four dimensions as in Table 3, and their scores show consistent agreement with GPT-4o's ratings (see **Appendix C.5** for detailed statistics). This suggests that GPT-4o is a reasonable proxy for expert judgment in our setting, while the core *Factual Alignment* metric already relies purely on expert-curated labels.

### 4.4 LLM RESOURCE CONSUMPTION (RQ4)

We compared the resource consumption of GMLLM against other LLMs, using the number of tokens consumed as the primary metric. The results are presented in Table 4. As the data shows, GMLLM demonstrates the lowest resource usage in all scenarios, and the margin of improvement is significant. In some cases, GMLLM achieves a reduction in computational cost by several orders of magnitude, which clearly demonstrates the superiority of our method. Beyond token usage, we also profile the cost of the GNN-Explainer step itself. On MalCP, optimizing the masks for a single project takes on average 10–11 seconds (see **Appendix E.1** for full statistics).

### 4.5 ABLATION STUDIES

#### 4.5.1 EFFECT OF RULE DESIGN AND MODEL COMPONENTS

Table 5: Ablation study of different rule configurations. **Bold** indicates the best performance.

| Configuration | Recall | Precision | Accuracy | Benign Recall |
|---|---|---|---|---|
| Structure-only (No $\mathcal{S}$) | 87.06 | 87.85 | 88.65 | 89.98 |
| Human Rules ($\mathcal{S}^{\text{comm}}$) | 90.43 | 91.53 | 91.86 | 93.04 |
| 50 Rules ($\mathcal{S}^{\text{comm}} \cup 0.5\mathcal{S}^{\text{data}}$) | 92.78 | 93.28 | 93.69 | 94.44 |
| Full GMLLM ($\mathcal{S}^{\text{comm}} \cup \mathcal{S}^{\text{data}}$) | **95.30** | **95.07** | **95.62** | **95.89** |

To understand where the gains of GMLLM come from, we first ablate the rule set $\mathcal{S}$ and the model components on MalCP (GPT-4o backend). Table 5 summarizes the effect of different rule configurations. "Structure-only" uses only structural graph features without any behavior rules. "Human Rules" ($\mathcal{S}^{\text{comm}}$) uses our manually designed common rules only. "50% Rules" randomly keeps half of the rules in $\mathcal{S}^{\text{data}}$ at inference time, and "Full GMLLM" uses the complete rule set. In all cases, $\mathcal{S}^{\text{comm}}$ and $\mathcal{S}^{\text{data}}$ are fixed; we do not regenerate rules from different fractions of the dataset.

The results indicate that behavior rules provide substantial but not exclusive benefits. Without any behavior rules, the graph-only variant still achieves 88.65% accuracy, showing that the structural representation itself is informative. Using only human-curated rules $\mathcal{S}^{\text{comm}}$ increases accuracy to 91.86%, already exceeding the GPT-4o direct baseline on MalCP (90.39%), which demonstrates that the framework does not rely solely on LLM-generated rules. Further enriching the rule set to $\mathcal{S}^{\text{comm}} \cup \mathcal{S}^{\text{data}}$ yields a monotonic improvement, with accuracy ultimately reaching 95.62%.

We also compare a GNN-only classifier with the LLM-only and full GMLLM variants on MalCP. The GPT-4o direct baseline from Table 2 attains 90.39% accuracy, the GNN-only model reaches 92.22%, and the full GMLLM attains 95.62%, corresponding to a 43.7% relative reduction in error rate (7.78% → 4.38%). This shows that (i) the GNN with rule-augmented graphs already forms a strong detector, and (ii) the LLM provides complementary gains on top of the GNN, particularly on harder cases.

Table 6: Effect of top-$K$ edge budget on detection performance and token usage on MalCP (GPT-4o). Note that token cost is reported specifically for the **Large** dataset bucket.

| $K$ (Max Edges) | Accuracy | Recall | Precision | Benign Recall | Token Cost |
|---|---|---|---|---|---|
| 10 | 92.97 | 94.18 | 90.75 | 91.96 | **435.45** |
| 20 | **95.62** | **95.30** | **95.07** | **95.89** | 640.06 |
| 30 | 95.19 | 95.45 | 94.03 | 94.96 | 810.33 |
| 50 | 94.88 | 95.99 | 92.94 | 93.97 | 1131.33 |

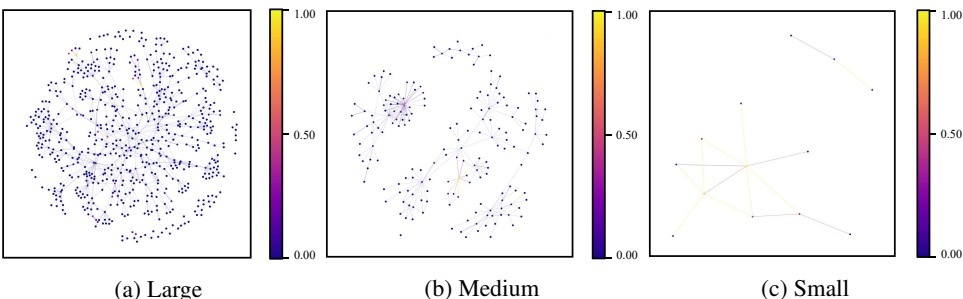

(a) Large  (b) Medium  (c) Small

Figure 3: A visualization of the code graph attention outputted with GMLLM when processing programs of different scales. In these graphs, the nodes and edges represent code elements and their corresponding relationships. The color intensity signifies the magnitude of attention, where colors closer to yellow indicate higher attention values.

### 4.5.2 SENSITIVITY TO THE ATTENTION-SUBGRAPH BUDGET $K$

As described in equation 11, the parameter $K$ controls the size of the attention subgraph that is passed to the LLM. In our implementation, we achieved this via a budgeted top-$K$ edge strategy: for each package, all call-graph edges are ranked by their explainer attention scores, and only the $K$ highest-scoring edges are retained to induce the subgraph. The same $K$ is shared across datasets.

We study the sensitivity of GMLLM to this budget on MalCP with GPT-4o by varying $K \in \{10, 20, 30, 50\}$ while keeping the GNN and explainer fixed. As shown in Table 6, increasing $K$ from 10 to 20 yields a clear performance gain (about +2.6 points in accuracy and +3.9 points in benign recall), suggesting that very small subgraphs may miss important context. For $K \geq 20$, however, detection metrics change only marginally, whereas the average token usage on Large packages grows almost linearly with $K$. We therefore fix $K = 20$ in all main experiments as a reasonable operating point that balances detection performance and token cost.

### 4.6 DEEPER INSIGHT

We visualized the graph attention generated by GMLLM, with the results presented in Figure 3. As illustrated, for the large-scale code, nearly all nodes are deep purple, which represents near-zero attention values. Only a small fraction of nodes and edges exhibit high attention, indicated by colors approaching yellow. This suggests that in larger-scale code projects, the proportion of code relevant to the malicious behavior is likely very small, which in turn explains the poor performance of LLMs on such samples. Our previous experiments corroborate this finding, underscoring the necessity of the additional attention mechanism in GMLLM. In contrast, on the small-scale dataset, the proportion of high-attention (yellow) nodes and edges increases significantly, and the performance of LLMs on these samples is correspondingly stronger.

## 5 CONCLUSION

We introduce GMLLM, a novel LLM-driven framework for malicious code detection designed to overcome a key limitation of LLMs: their susceptibility to noise from irrelevant code in large projects. Our approach leverages a lightweight GNN, trained on easily accessible data, which is interpreted during inference to produce attentions. These attentions direct the LLM to focus on potentially malicious code segments. As a result, GMLLM outperforms current state-of-the-art tools, rendering the direct use of LLMs for malicious code detection a practical possibility in real-world settings.

## REPRODUCIBILITY STATEMENT

Our experiments provide both data and code to ensure reproducibility. These resources are included in the supplementary materials, with further details available in the accompanying README.md file.

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

## A  USAGE OF LARGE LANGUAGE MODEL

In our paper, we used LLMs to assist with polishing the writing, including correcting grammatical errors and making the sentences more consistent with academic English writing conventions.

## B  EXTENDED RELATED WORKS

### B.1  MALICIOUS CODE DETECTION METHODS

Malicious code detection is crucial for protecting information security, preventing data leaks, and defending against cyberattacks. It effectively prevents system damage and the loss of sensitive information. Conventional malicious code detection can be divided into three main categories: metadata-based malicious code detection, rule-based malicious code detection, and learning-based malicious code detection.

Metadata-based malicious code detection relies on discovering malicious code through information such as the package name and attributes. For example, Neupane et al. (Neupane et al., 2023) and Taylor et al. (Taylor et al., 2020) use package names to identify malicious code. This approach, however, is relatively weak, as attackers can easily bypass it by mimicking the names of legitimate code packages (Kaplan & Qian, 2021). Existing work (Vu et al., 2020) attempt to optimize malicious code detection by calculating the edit distance between package names, but this may lead to a high false positive rate.

In contrast to the aforementioned methods, rule-based approaches are more detailed but require experts to manually design the rules, making them more resource-intensive. Specifically, there are existing rule-based detection tools for interpreted languages, including Yara (VirusTotal, 2023), Bandit (PyCQA, 2023), OSSGadget (Microsoft, 2023), and OSSF (OSSF, 2023). These tools are highly efficient. Furthermore, Duan et al. (Duan et al., 2021) and Huang et al. (Huang et al., 2024) have combined static and dynamic analysis with related rules for malicious code detection, but these methods also rely on specialized knowledge and custom rules.

Recently, with the rise of deep learning techniques, learning-based malicious code detection methods have become increasingly popular. Various deep learning and machine learning methods have been employed for malicious code detection. Among them, Yadav et al. (Yadav et al., 2022) converts bytecode into images and uses convolutional neural networks (CNN) for classification, achieving high detection accuracy. They also applies a pseudo-label stacked autoencoder for semi-supervised learning, improving the model's generalization capability. At the same time, many studies focus on extracting different types of features (such as API calls, permissions, system calls, opcode, etc.) for malicious code detection, thus expanding the application range of learning-based methods. Muhammad et al. (Aamir et al., 2024) extracts opcode frequencies, API calls, and permissions, transforms them into two-dimensional images, and employs convolutional neural networks (CNN) for malicious code detection. Wu et al. (Wu et al., 2023) utilizes text mining methods to extract key features from application code, followed by the generation of call graphs and the integration of Bi-LSTM and GNN for analysis. Many other studies focus on enhancing the robustness of malicious code detection models against adversarial attacks. Yumlembam et al. (Yumlembam et al., 2023) combines variational graph autoencoders (VGAE) with generative adversarial networks (GAN) to strengthen the model's robustness when attackers are aware of the model's features. Amin et al. (Amin et al., 2022) uses long short-term memory generative adversarial networks (LSTM-GAN) to process opcode sequence features, improving the model's ability to detect adversarial attacks. The methods mentioned above are all based on learning from malicious code data and constructing models. Li et al. (Li et al., 2024), on the other hand, approaches the problem from the perspective of the attacker, successfully allowing malicious code variants to persist in machine learning-based, signature-based, and hybrid anti-malicious code software. Li et al. (Li et al., 2024) utilizes the uncertainty estimation from the corrected model output to adjust the prediction results, thereby enhancing the accuracy of the DNN model. As for methods related to the recently popular large language models, we will introduce them in the next part.

### B.2  MALICIOUS CODE DETECTION WITHIN PYPI

The widespread use of open-source software has made PyPI vulnerable to significant security challenges, with malicious code attacks leading to global disruptions and billions of dollars in losses. Okafor et al. (Okafor et al., 2022) highlighted issues such as service interruptions and cybersecurity risks. Dambra et al. (Dambra et al., 2023) noted that many studies have proposed machine learning models for malicious code detection, achieving near-perfect performance, but with differing methodologies and feature extraction techniques.

Static code analysis is a key technique for detecting malicious Python packages. Gobbi and Kinder (Gobbi & Kinder, 2023) proposed using the CodeQL framework for detecting malicious npm packages, successfully identifying 125 malicious packages with no false positives. Liu et al. (Liu et al., 2024) discussed challenges posed by machine learning-based malicious code detection that relies on binary files, presenting new opportunities for static analysis. Dambra et al. (Dambra et al., 2023) collected the largest balanced dataset for

malicious code detection, revealing that static features outperform dynamic ones, and that larger datasets and more samples per family improve accuracy. Arp et al. (Arp et al., 2022) emphasized the importance of dataset selection and evaluation in applying machine learning to security. Vu et al. (Vu et al., 2023) conduct study that reveal repository administrators require extremely low false positive rates (below 0.1%). They found that a socio-technical malicious code detection system has emerged, where external security researchers scan for malicious code, filter the results, and report to administrators. MPHunter (Liang et al., 2023) and Ea4mp (Sun et al., 2024) introduced language model into malicious code detection.

Malicious code detection within Python packages faces challenges such as anti-detection techniques like obfuscation, limited review capabilities of maintainers, and high false positive rates. Future research should focus on hybrid detection methods combining static analysis, machine learning, and behavior analysis, alongside advancing techniques to counteract code obfuscation and exploring the use of large language models in detection.

## B.3 LLM FOR MALICIOUS CODE DETECTION

With the rapid development of Large Language Models (LLMs), their application in the field of malicious code detection, particularly in code analysis, behavior analysis, data augmentation, malicious code generation, and integrated frameworks, has garnered widespread attention.

LLMs leverage their powerful code comprehension capabilities to analyze source code or decompiled code to identify potential malicious behaviors. Research by Fang et al. (Fang et al., 2024) indicates that while LLMs perform well in certain code analysis tasks, their performance significantly declines when dealing with complex or obfuscated code. The Maltracker method proposed by Yu et al. (Yu et al., 2024), which combines LLMs with traditional static analysis techniques, significantly improves the accuracy of malicious code detection. In contrast to static code analysis, behavior-based methods focus on the dynamic behavior of malicious code during execution. Akinsowon and Jiang (Akinsowon & Jiang, 2024) demonstrate that LLMs can significantly improve detection rates by analyzing the dynamic behavioral characteristics of malicious code, especially for those malicious code samples that are difficult to detect through static analysis.

The generative capabilities of LLMs provide solutions to the issue of insufficient malicious code samples. Yu et al. (Yu et al., 2024) extend the diversity and representativeness of datasets by translating malicious functions into JavaScript, thus enhancing the accuracy of malicious code detection. The powerful generative ability of LLMs can both enhance security defenses and potentially be misused to generate more sophisticated malicious software. Khan et al. (Khan et al., 2024) propose a risk mitigation framework aimed at preventing malicious code through the use of LLMs.

Given the complexity of evolving malicious code threats, single technologies are often inadequate. The integrated framework proposed by Khan et al. (Khan et al., 2024) combines the advantages of LLMs with traditional security techniques, designing a multi-layered, multi-module detection system that effectively responds to constantly changing malicious code threats.

## B.4 DISTINCTIONS FROM CODE EMBEDDING AND DYNAMIC ANALYSIS METHODS

Beyond metadata- and rule-based detectors, recent work has explored pre-trained code models and dynamic traffic analysis for software security. However, these approaches differ from our setting in both granularity and analysis paradigm.

Several recent methods build security detectors on top of CodeBERT. For example, ugSliceVul (Zou et al., 2025) and VulCoBERT (Xia et al., 2024) utilize pre-trained models for function-level vulnerability detection, and typically operate on short C/C++ snippets. Likewise, the Multiclass Software Defect Prediction (MSDP) model by Hussain et al. (Hussain et al., 2025) and VulD-CodeBERT by Xiong and Dong (Xiong & Dong, 2024) demonstrate that fine-tuned CodeBERT can effectively classify diverse defects and vulnerabilities at the snippet or file level. In contrast, malicious Python packages in our setting often rely on project-wide logic spanning multiple modules and installation hooks (e.g., setup.py). The standard context window of these models (e.g., 512 tokens for CodeBERT (Feng et al., 2020)) forces heavy truncation when applied to entire packages with thousands of tokens, making it difficult to capture long-range, cross-file interactions. Although some studies apply CodeBERT to Python source snippets (Zhao et al., 2024), to the best of our knowledge they do not release reproducible, end-to-end systems for package-level scanning. For this reason, we adopt MPHunter and Ea4mp as our primary baselines, as they are specifically tailored to package-level Python/PyPI malware.

Orthogonal to static code analysis, MalPaCA (Nadeem et al., 2019) focuses on clustering malware based on network traffic traces captured during sandboxed execution. This represents a dynamic analysis paradigm that requires executing each sample and collecting PCAP logs. Our GMLLM framework, by contrast, targets *static* supply-chain scanning: it operates directly on Python source code and dependency/call graphs, aiming to detect threats before installation or execution. Using MalPaCA as a baseline in our setting would require dynamically

executing thousands of packages to generate traffic traces, which is impractical at PyPI scale and conceptually distinct from our static-analysis objective.

## C  IMPLEMENTATION DETAILS

### C.1  GRAPH CONSTRUCTION DETAILS

We begin the graph construction by extracting the dependencies within the code. We traverse all '.py' files in the project directory and parse the source code of each file into an AST object, constructing the set $\mathcal{V}^{\text{code}}$ based on the nodes in the AST. The $\mathcal{V}^{\text{code}}$ includes class and function nodes within the AST, and each Python file itself is treated as a "module" type node. The source code corresponding to each node will be stored as a node attribute.

The edge set $\mathcal{E}^{\text{code}}$ is divided into dependency edges and call relationship edges. The dependency edges include definition edges, inheritance edges, and decorator edges, which are obtained through AST parsing. These edges are derived as follows:

1. **Definition Edge**: When a definition is encountered within a scope, a definition edge is created from the current scope's node to the newly defined class or function node.

2. **Inheritance Edge**: When processing class definition nodes, we check the base classes and create an inheritance edge from the current class node to each of its base class nodes.

3. **Decorator Edge**: We iterate over the decorator list for class and function definitions. By parsing the decorator expressions, it creates a decorator edge from the function node representing the decorator to the node representing the decorated class or function.

Next, we outline the call relationship edges. We traverse the AST again to identify and record these relationships. We specifically target the AST nodes representing function calls to identify function invocations. To accurately resolve the identity of the called function, the analyzer combines various pieces of information, including the import mapping, aliases, and the current scope context. Once the function is identified, an edge is created from the calling node to the called function node. These call relationships are recorded as different types of edges based on the context, including:

1. **Function-level Call**: If the invocation occurs within another function or method, a Function-level Call edge is recorded.

2. **Module-level Call**: If the invocation occurs at the module level (i.e., outside any function or class), a Module-level Call edge is recorded.

3. **Hook**: We specifically handles calls to the 'setup()' function or 'setuptools.setup()' function. We check the 'cmdclass' parameter, and if a custom class overrides the installation command ('install'), a hook edge is created from the 'setup()' node to the 'run()' method of the custom class, capturing this potentially high-risk installation-time code execution behavior.

### C.2  RULE GENERATION DETAILS

As for the construction of the each rule $s$ within $\mathcal{S}$, we used Python's lambda expressions and employed a functional programming approach to dynamically match different code behaviors. Each rule determines whether a behavior matches a malicious characteristic based on function names, method calls, or module names. The following presents two examples:

```
Examples of Sensitive Behavior Rules

"network":  lambda n:  n.startswith(("socket.", "requests.",
"urllib.")),
"phishing":  lambda n:  n in ("requests.post", "HTTPConnection")
```

The 'network' rule checks if network-related libraries (such as socket, requests, urllib) are being invoked, while the 'phishing' rule focuses on HTTP request methods potentially related to phishing attacks. The rules are defined as conditional functions through lambda expressions, returning True when specific conditions are met, indicating that the behavior matches the malicious characteristic. Compared to other approaches, we found that the automatic generation based on LLM (large language models) is more suitable for this simple and explicit rule construction pattern.

The prompt that is used to generate $\mathcal{S}^{\text{comm}}$ is as follows:

**Prompt $\mathcal{S}^{\text{comm}}$**

{ "role": "system",
"content": """"You are an AI model designed to identify and analyze sensitive behaviors in Python programming code. Sensitive behaviors refer to actions or patterns in the code that could result in security vulnerabilities, data privacy violations, or unintended exposure of sensitive information. These behaviors include but are not limited to insecure network operations, improper handling of user inputs, data encryption weaknesses, and any practices that may inadvertently expose sensitive data or increase the risk of attacks.

Please provide a list of potential sensitive behaviors based upon common Python coding and analysis knowledge. For each sensitive behavior, provide a brief explanation of why it is considered sensitive and how it could lead to vulnerabilities or privacy violations. Additionally, provide a Python function (using lambda expressions and a functional programming approach) to detect and match the corresponding behavior. The function should analyze elements like function names, method calls, variable names, or imported modules.

The rule for each sensitive behavior should determine whether a particular piece of code exhibits a malicious or risky characteristic based on its structure, such as the use of certain function names, method calls, or external libraries. Here are some examples:

Example 1: Detecting Network Operations:

"network": lambda n: n.startswith("socket.", "requests.", "urllib."),

......

Each rule should be a lambda expression that matches specific patterns, function calls, or method names relevant to the corresponding sensitive behavior."""" }

The prompt that is used to generate $\mathcal{S}^{\text{data}}$ is as follows:

**Prompt $\mathcal{S}^{\text{data}}$**

{ "role": "system",
"content": """"You are an AI model designed to identify and analyze sensitive behaviors in Python programming code. Sensitive behaviors refer to actions or patterns in the code that could result in security vulnerabilities, data privacy violations, or unintended exposure of sensitive information. These behaviors include but are not limited to insecure network operations, improper handling of user inputs, data encryption weaknesses, and any practices that may inadvertently expose sensitive data or increase the risk of attacks.

You are provided with a Python code snippet that may contain sensitive behaviors or security vulnerabilities. Please analyze the code and summarize, and provide a list of potential sensitive behaviors detected in the provided Python code. For each sensitive behavior, provide a brief explanation of why it is considered sensitive and how it could lead to vulnerabilities or privacy violations. Additionally, provide a Python function (using lambda expressions and a functional programming approach) to detect and match the corresponding behavior. The function should analyze elements like function names, method calls, variable names, or imported modules.

The rule for each sensitive behavior should determine whether a particular piece of code exhibits a malicious or risky characteristic based on its structure, such as the use of certain function names, method calls, or external libraries. Here are some examples:

Example 1: Detecting Network Operations:

"network": lambda n: n.startswith("socket.", "requests.", "urllib."),

......

Each rule should be a lambda expression that matches specific patterns, function calls, or method names relevant to the corresponding sensitive behavior."""" }
{"role": "user", "content": $< X_i >$}

## C.3  PROMPT $\rho^{\text{ANA}}$

The detailed content concerning $\rho^{\text{ana}}$ is:

---

**Prompt $\rho^{\text{ana}}$**

{ "role": "system",
"content": """"You are a PyPI package security auditor. You have been provided with the 'high-attention subgraph' structure of a PyPI package script, which only includes node names and call relationships. Based solely on this subgraph structure, please answer the following: Is this structure indicative of potential malicious activity? (Respond only with 'Malicious' or 'Benign')
Provide your reasoning. Be cautious not to label a package as malicious based on a single suspicious behavior without considering the broader context of the entire report. If mitigation is needed, identify the highest priority nodes or calls for input validation or permission checks. Response Format:
Verdict:
Reasoning:
Mitigation:"""},
{"role": "user", "content": $< \textbf{Att}(G_j^{\textbf{code}}) >$}

---

## C.4 DETAILS CONCERNING EVALUATION UPON MALICIOUS BEHAVIOR DESCRIPTION

The evaluation was conducted exclusively on samples drawn from the malicious subset of the MalCP dataset, each of which includes a verified ground-truth behavior summary from Synk (Snyk, 2024).

We selected GPT-4o as the evaluator due to its superior reasoning and instruction-following capabilities among all models tested in our experiments. Each explanation-ground truth pair was fed into the model using the same prompt template, and results were collected via automated scripts.

For models that misclassified these malicious samples as benign, a zero score was assigned across all evaluation dimensions as a penalty. This penalty mechanism is justified as models that fail to recognize malicious intent inherently cannot provide meaningful behavioral explanations. The full prompt template is shown below.

**Prompt** $\rho^{\text{judge}}$

{ "role": "system",
"content": """"You are a senior cybersecurity analyst with deep expertise in threat intelligence. Your role is to evaluate an AI-generated explanation of malicious behavior by its depth of insight, not just its factual accuracy.
Your assessment has two parts:
Part 1: Intrinsic Quality Evaluate the explanation's internal strength—without referencing the ground truth—using the following criteria:
1. Threat Tactic Generalization (1–5)
How well does it generalize from specific functions or code elements to a recognized cybersecurity tactic (e.g., 'Reconnaissance', 'Defense Evasion', 'Exfiltration')?
→ 5: Clearly identifies and frames the analysis using a standard attack pattern (e.g., MITRE ATT&CK).
→ 4: Identifies a general malicious purpose (e.g., data collection), but lacks formal tactical framing.
→ 3: Recognizes suspicious behavior but generalizes weakly or vaguely (e.g., "could be malicious").
→ 2: Lists actions with minimal abstraction (e.g., "gets username → collects info").
→ 1: Only describes concrete functions with no generalization.
2. Execution Path Traceability (1–5)
How clearly does it reconstruct a step-by-step execution flow, including function calls, data movement, and control logic?
→ 5: Presents a complete, logical, and verifiable sequence of actions.
→ 4: Shows a mostly coherent flow but omits minor steps or data dependencies.
→ 3: Outlines key stages (e.g., collect → encode → send), but with gaps or unclear transitions.
→ 2: Mentions multiple functions without clear order or causality.
→ 1: No discernible execution path.
3. Evidence Groundedness (1–5)
Are claims directly and tightly supported by specific code elements (e.g., function calls, strings, variables)?
→ 5: Every significant claim is explicitly tied to observable code.
→ 4: Most claims are evidence-backed; minor inferences are reasonable.
→ 3: Core behaviors are supported, but some conclusions extend beyond direct evidence.
→ 2: Sparse or generic references to code; many unsupported assertions.
→ 1: Claims are vague, speculative, or entirely disconnected from code.
Part 2: Factual Alignment
Compare the explanation to the Ground Truth Summary:
4. Factual Alignment (1-5)
Does it correctly identify and accurately describe the primary malicious behavior?
→ 5: Accurate and complete—captures the core behavior and key details.
→ 3: Partially correct—identifies the general type of malicious activity but misses or misrepresents critical elements.
→ 1: Fundamentally wrong or entirely misses the core behavior.
OUTPUT (JSON):
{
   "quality_scores": {
   "threat_tactic_generalization": int,
   "execution_path_traceability": int,
   "evidence_groundedness": int
},
"alignment_score": {
   "factual_alignment": int
} } """"},
{"role": "user", "content": Explanation_To_Evaluate:    "{explanation}"
Ground_Truth_Summary:    "{ground_truth}"}

The model returns structured JSON output, including both quantitative scores and a textual justification. To ensure response validity, all outputs were automatically checked for schema compliance (valid JSON). If an output was malformed, we re-issued the query up to 3 times. Only consistently valid outputs were included in final scoring; invalid ones were excluded from aggregate metrics. A concrete example is given below:

```
LLM-as-a-Judge Evaluation Examples

"quality_scores":
{"threat_tactic_generalization":  4, "execution_path_traceability":
3, "evidence_groundedness":  3},
"alignment_score":  {"factual_alignment":  3},
"Reasoning":  "The provided Python code is a sophisticated malware
that performs a variety of malicious activities.  It is designed
to steal sensitive information such as passwords, cookies, and
tokens from various applications and browsers.  The code includes
functionality to decrypt encrypted data, exfiltrate data to a remote
server, and manipulate clipboard data to replace cryptocurrency
wallet addresses.  It also attempts to evade detection by checking
for virtual machine environments and debugging tools.  Additionally,
the code includes a mechanism to persistently install itself on
the victim's system by adding itself to the startup programs.  The
use of obfuscation and deceptive descriptions further indicates
malicious intent.  The code interacts with external servers using
hardcoded URLs, which are used to send stolen data and receive
commands.  This behavior is typical of malware designed for data
theft and system compromise."
```

### C.5 HUMAN VALIDATION OF EXPLANATION-QUALITY METRICS

To validate the reliability of using GPT-4o as an automatic evaluator for the explanation-quality metrics in Table 3, we conducted a human study with two security practitioners.

We randomly sampled 70 model-generated explanations from MalCP. For each explanation, the two experts independently scored four dimensions using the same discrete scale as in the main paper: *Threat Tactic Generalization*, *Execution Path Traceability*, *Evidence Groundedness*, and *Factual Alignment*. We then computed (i) inter-annotator agreement between the two experts and (ii) the correlation between GPT-4o's scores and the mean of the two human scores.

Table 7 reports Spearman's rank correlation coefficients ($\rho$) for both Human–Human and GPT-4o–Human comparisons. Human–Human agreement is consistently moderate to strong ($\rho \in [0.61, 0.81]$, $p < 10^{-7}$), which is in line with the natural variability expected for this kind of subjective, multi-dimensional task. GPT-4o's scores exhibit similarly strong correlations with the human averages ($\rho \in [0.72, 0.87]$, $p < 10^{-12}$) across all four dimensions, in some cases slightly higher than Human–Human.

Table 7: Spearman's $\rho$ (and $p$-values) for Human–Human agreement and GPT-4o–Human correlation on the four explanation-quality dimensions.

| Evaluation Dimension | $\rho$ (Human–Human) | p-value (Human–Human) | $\rho$ (GPT–Human) | p-value (GPT–Human) |
|---|---|---|---|---|
| Threat Generalization | 0.613 | $1.71 \times 10^{-8}$ | 0.820 | $3.31 \times 10^{-18}$ |
| Execution Path Traceability | 0.809 | $2.05 \times 10^{-17}$ | 0.870 | $1.41 \times 10^{-22}$ |
| Evidence Groundedness | 0.755 | $3.93 \times 10^{-14}$ | 0.758 | $3.02 \times 10^{-14}$ |
| Factual Alignment | 0.708 | $6.65 \times 10^{-12}$ | 0.724 | $1.34 \times 10^{-12}$ |

Overall, these results support the use of GPT-4o as a reasonable proxy for expert judgment when evaluating explanation quality, while the core *Factual Alignment* metric in the main paper already relies directly on expert-curated malware analysis reports as ground truth.

## D EXPERIMENT DETAILS

### D.1 DATASETS

To evaluate our method under real-world conditions, we constructed a benchmark dataset named **MalCP (Malicious Codes from PyPI)**. This dataset integrates malicious packages from three public sources, aligns their behavioral labels using the Snyk advisory system (Snyk, 2024), and filters them using size-based heuristics to ensure reproducibility and label consistency.

### D.1.1 DATA AGGREGATION FROM PUBLIC SOURCES

To construct the MalCP dataset, we compiled malicious Python packages from the PyPI ecosystem by integrating multiple open-source intelligence sources, including Mal_OSS (Guo et al., 2023), Backstabbers Knife Collection (Ohm et al., 2020), and Datadog's open malware dataset (Guarddog, 2023). This resulted in a large and diverse collection of real-world malicious packages.

While the source datasets provide access to source code and limited metadata (such as upload timestamps or package descriptions), the vast majority do not contain structured or human-readable descriptions of malicious behaviors. To enable behavior-aware detection and analysis, we retrieved and matched advisory content from the **Snyk Vulnerability Database**.

Specifically, for each package in our initial collection, we attempted to crawl the `overview` section from the Snyk Vulnerability Database, which typically contains a concise, high-level description of the malicious activity. Successful matches resulted in 7,377 annotated packages. These overviews were stored in a structured `vuln_report.json` file for each sample. Examples include:

> "10Cent11 is a malicious package. It creates a reverse shell to a hard-coded IP address, giving the attacker full control over an infected system."

This standardization ensures that all labels are derived from a consistent and widely trusted security knowledge base.

### D.1.2 SIZE-BASED FILTERING CRITERIA

To ensure the analytical value and structural completeness of the dataset, we applied a **size-based filtering policy** that eliminated trivial or non-informative samples while preserving diversity across package sizes. The thresholds of **5KB** and **10KB** were empirically determined by inspecting the distribution of raw package sizes:

- **Large** Packages larger than 10KB: Fully retained.

- **Medium** Packages between 5KB and 10KB: Partially sampled. We randomly sampled a representative subset to maintain approximate proportionality with the large-size group. Sampling was performed using a fixed random seed for reproducibility.

- **Small** Packages smaller than 5KB: A small, balanced subset. We retained a small, behaviorally diverse subset. At least one sample was selected per behavior category (e.g., reverse shell, credential theft), ensuring semantic diversity despite the size limitation.

The distribution of the final 1,659 packages by size category is shown in Table 8.

Table 8: Final Samples by Package Size

| Package Size Range | Samples Retained | Selection Criteria |
|---|---|---|
| Large(> 10 KB) | 461 | All retained |
| Medium(5–10 KB) | 588 | Partial sampling |
| Small(< 5 KB) | 610 | Balanced selection |
| **Total** | 1,659 | – |

### D.1.3 ATTACK-TYPE DISTRIBUTION IN MALCP

For each malicious package in MalCP, we annotate its primary attack type by combining (1) the human-written analysis reports from Snyk (used as ground-truth labels) and (2) the high-level tactics in the MITRE ATT&CK framework Strom et al. (2018). We collapse the original fine-grained descriptions into eight coarse-grained categories: Execution, Impact, Credential Access / Collection, Defense Evasion, Exfiltration, Command and Control, Persistence, and Other (for rare or mixed behaviors).

Table 9 summarizes the resulting distribution. Execution covers cases where the package directly launches or injects a payload (e.g., malicious installation hooks); Impact captures destructive or resource-abuse behaviors (e.g., wiping files, crypto-mining); Credential Access / Collection includes stealing API keys, SSH keys or browser credentials; the remaining categories correspond to evasion, data exfiltration, command-and-control channels, and long-lived persistence mechanisms.

As the table shows, the three most common tactics—Execution (33.8%), Impact (23.1%), and Credential Access / Collection (23.0%)—each account for roughly one third of malicious samples, while the remaining

tactics form a non-trivial tail (Defense Evasion 8.0%, Exfiltration 6.4%, Command and Control 2.5%, Persistence 0.2%, Other 3.0%). This reflects realistic attack behavior observed in the wild rather than an artificially balanced benchmark, and ensures that MalCP exercises detectors across a broad range of malicious intents.

Table 9: Distribution of Malicious Samples across MITRE ATT&CK Tactics

| Attack Type | Count | Percentage |
|---|---|---|
| Execution | 561 | 33.82 |
| Impact | 383 | 23.09 |
| Credential Access / Collection | 382 | 23.03 |
| Defense Evasion | 132 | 7.96 |
| Exfiltration | 106 | 6.39 |
| Other | 49 | 2.95 |
| Command and Control | 42 | 2.53 |
| Persistence | 4 | 0.24 |
| Total | 1659 | 100 |

### D.1.4  DATASET FILTERING PIPELINE

The construction of the MalCP dataset followed a three-stage process:

1. **Aggregation:** Malicious packages were consolidated from the PyPI ecosystem using multiple open-source intelligence sources.

2. **Behavioral Annotation:** Each package was enriched with a structured behavior summary derived from the Snyk advisory system.

3. **Filtering:** Samples were filtered based on archive structure and size constraints to ensure semantic richness and practical utility.

### D.2  SETTINGS

This section outlines the end-to-end pipeline used in our experiments, including graph preparation, model training, explanation generation, and runtime environment setup.

### D.2.1  GRAPH PREPARATION AND LABELING

Each package is parsed into a static code graph as described in **Appendix** C.1. Nodes represent program elements (e.g., functions, classes, literals, or modules) and are enriched with metadata such as identifier name, AST type, and file-level context. Edges capture both internal control flow (e.g., call relationships) and external dependencies (e.g., inheritance or decoration).

Each node feature vector consists of:

- a 64-dimensional embedding for the identifier name;

- a 16-dimensional embedding of the AST type;

- a one-hot vector representing malicious behavior labels.

Behavior labels are inferred from a synthesized rule set generated by prompting an LLM with structured behavior definitions. Each rule is a restricted lambda expression compiled via an AST whitelist. If rule synthesis fails, a fallback static rule set is used. Behavior features are stored as node attributes and aligned to a fixed `behavior2idx` vocabulary.

The graphs are then serialized into PyTorch Geometric-compatible tensors using frozen vocabularies (`name2idx`, `type2idx`, `edge_type2idx`, `behavior2idx`) constructed from the training set and reused across all runs.

### D.2.2  GRAPH CLASSIFICATION (FOR EXPLANATION ONLY)

To extract semantically meaningful subgraphs for explanation, we train a lightweight graph neural classifier over the prepared graphs. The model is not used for standalone detection.

We adopt a two-layer GCN architecture, where each node's input is the concatenated feature vector described above. Each GCN layer is followed by a ReLU activation and dropout (rate = 0.6). A global mean pooling layer aggregates node embeddings to the graph level, which is then fed into a linear classifier with softmax output to predict package-level maliciousness.

Training is conducted using an 80/20 train/test split at the package level with a fixed seed. No validation set is used. We use the Adam optimizer (learning rate $= 10^{-3}$, weight decay $= 10^{-3}$), a batch size of 128, and 100 training epochs. Early stopping may be applied based on test F1 score.

### D.2.3 EXPLANATION AND LLM REVIEW

To support human-interpretable analysis, we apply `GNNExplainer` on the trained graph classifier to extract salient subgraphs that contribute most to the prediction. Based on the learned importance mask, we extract a subset of high-salience edges and their associated context.

These extracted subgraphs are submitted to a large language model for semantic judgment. Each LLM response returns a structured output containing a verdict, reasoning, and possible mitigation. This procedure allows us to incorporate symbolic reasoning and external knowledge into the evaluation of structural malicious patterns.

Our framework supports different LLM backends in practice, but we do not depend on any particular model configuration for core results.

### D.2.4 RUNTIME ENVIRONMENT AND REPRODUCIBILITY

Our experiments are conducted on standard research hardware with NVIDIA GPUs ($\geq$16GB VRAM). All code is implemented in Python 3.7+, using the following core libraries:

- PyTorch 1.13.1 with CUDA 11.7
- PyTorch Geometric 2.3.1
- Scikit-learn 1.0.2
- OpenAI SDK $\geq$1.39.0

We fix random seeds across Python, NumPy, and PyTorch for reproducibility. Static vocabularies, synthesized rules, and all critical dependencies are released as part of our codebase. CUDA full determinism is enabled where applicable.

## E EXTRA EXPERIMENTAL RESULTS

### E.1 RUNTIME OF THE GNN-EXPLAINER STEP

We profile the cost of the per-package mask optimization used to extract attention subgraphs (Sec. 3.2). The driver script logs wall-clock time from launching `explainer_main.py` to its termination. All experiments run on a single NVIDIA GPU (CUDA 11.7, cuDNN 8.5) with a fixed budget of 100 gradient steps per package.

Table 10 summarizes the runtime distribution on MalCP (Small/Medium/Large buckets) and on the Datadog dataset. Across all subsets, the mean per-package runtime is around 10–11 seconds, the median is close to 9 seconds, and the 95th percentile ($P_{95}$) stays below roughly 20 seconds. The similar numbers across size buckets are largely due to a substantial constant overhead (Python startup, checkpoint loading, graph preparation), together with the fixed 100-step optimization budget; the part of the computation that scales with graph size is not dominant at this scale.

Table 10: Runtime statistics of the GNN-Explainer. "Small/Medium/Large" are the MalCP size buckets; Datadog is one of the public datasets used in the main experiments.

| Subset / Dataset | Mean $\pm$ Std (s) | Median (s) | Min / Max (s) | $P_{95}$ (s) | Exceeded $P_{95}$ (%) |
|---|---|---|---|---|---|
| Small | $10.18 \pm 4.10$ | 8.95 | 6.96 / 25.31 | 18.32 | 5.09 |
| Medium | $10.59 \pm 3.92$ | 8.96 | 7.08 / 25.47 | 18.57 | 5.05 |
| Large | $10.73 \pm 4.70$ | 8.86 | 6.83 / 55.28 | 19.16 | 5.04 |
| Datadog | $10.76 \pm 4.21$ | 8.68 | 7.15 / 22.86 | 20.45 | 5.04 |

Overall, the GNN-Explainer step adds roughly 10 seconds of offline processing per package under our fixed 100-step budget, with about 95% of packages finishing within 18–20 seconds across different datasets. Since GMLLM is intended for *offline batch scanning* of new releases, rather than interactive request-time serving, this level of overhead is acceptable, especially when combined with the substantial reduction in LLM token usage reported in Sec. 4.4. In practice, the explainer step budget (currently 100) can be reduced (e.g., to 30–50 steps) or explanations can be cached across scans to further reduce latency.

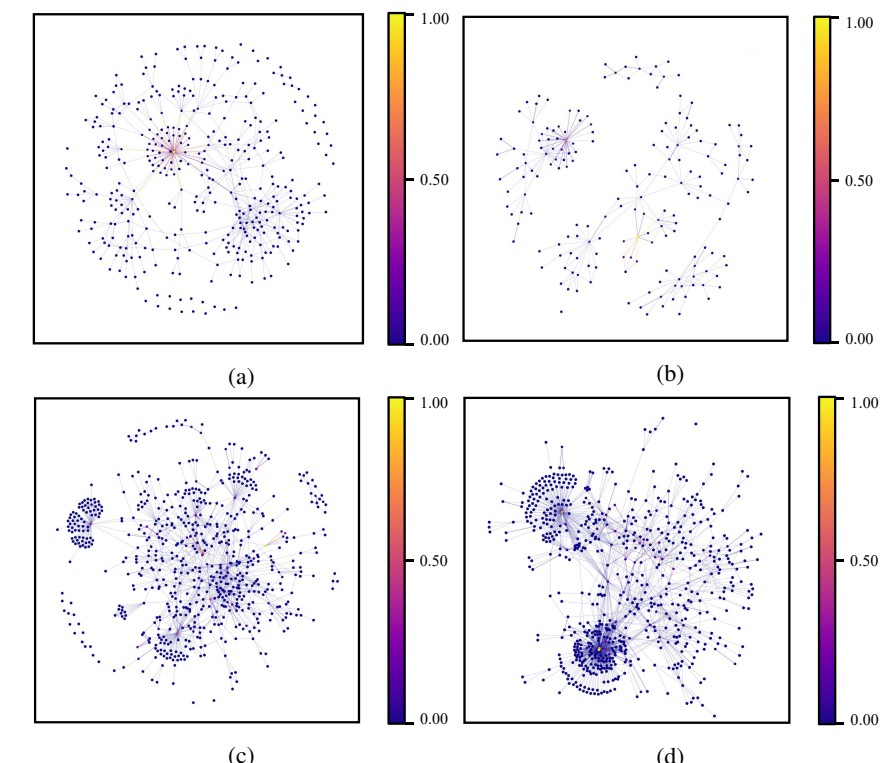

Figure 4: Visualizations of the code graph attention outputted with GMLLM when processing programs of large scales.

## E.2 EXTRA VISUALIZATION RESULTS

Figures 4, 5, and 6 provide additional visualizations of the code graph attention. From the figures, it can be observed that malicious code nodes or relationships, i.e., those nodes and edges with colors closer to yellow, often exhibit positions with a higher number of neighboring nodes or more complex associations. This topological property is quite evident and also suggests that graph representation learning is effective for the preliminary identification of these nodes.

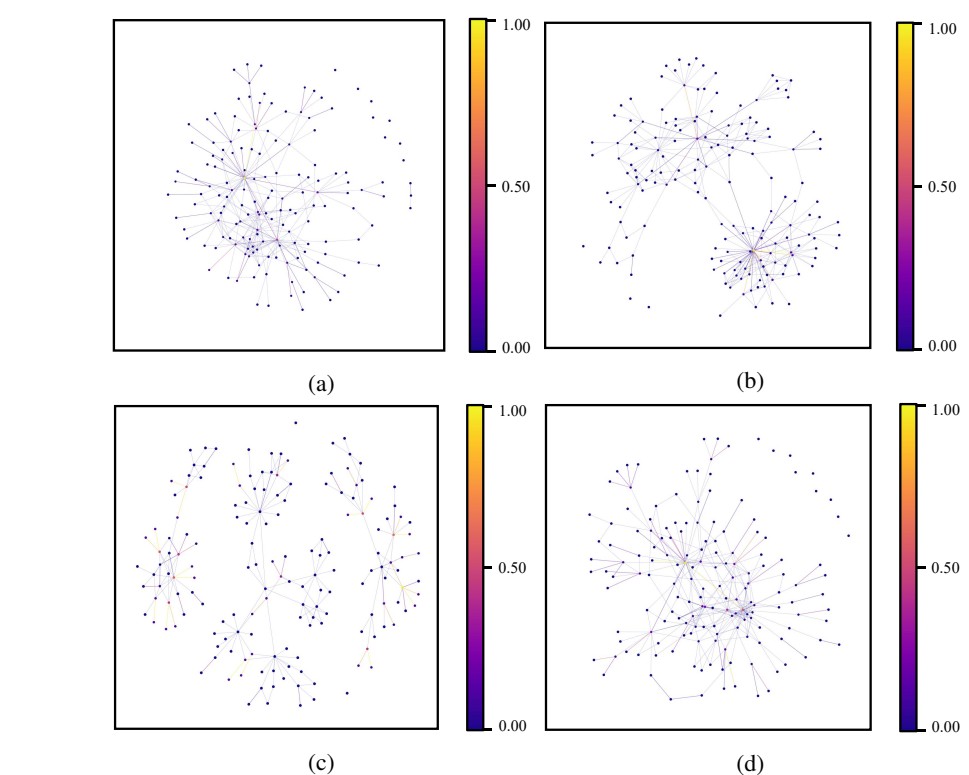

Figure 5: Visualizations of the code graph attention outputted with GMLLM when processing programs of medium scales.

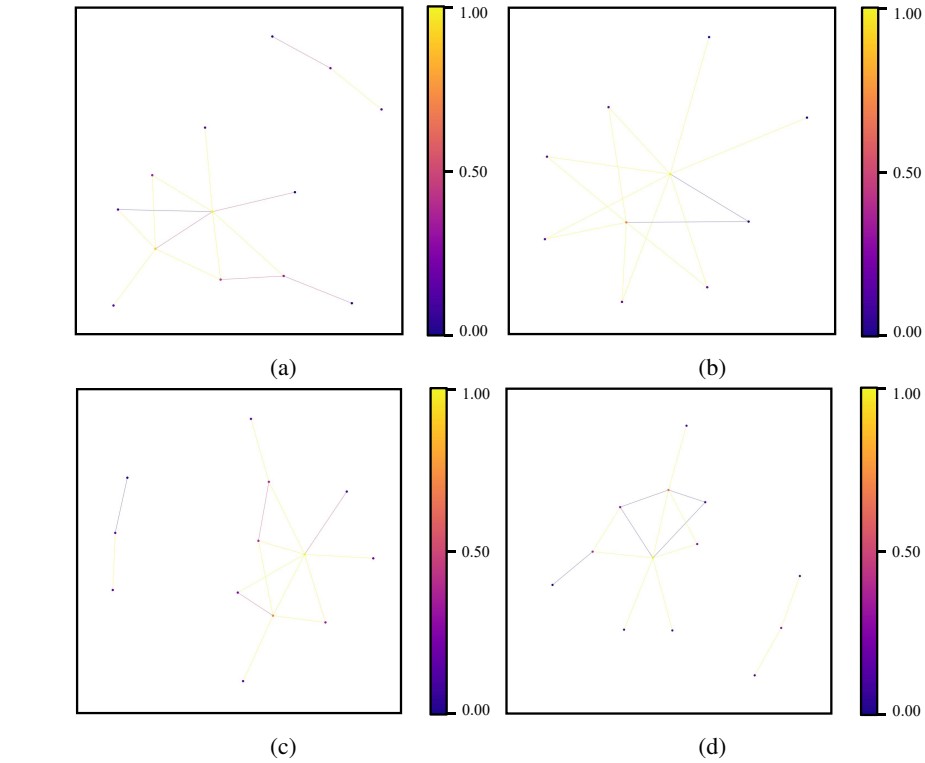

Figure 6: Visualizations of the code graph attention outputted with GMLLM when processing programs of small scales.

### E.3 Sample Model Judgments from High-Attention Subgraphs

To further illustrate the security reasoning capability of our approach, we present illustrations of outputs of GMLLM.

The following examples were generated by GMLLM based on ChatGPT 4o. For each case, the model produced a binary verdict ('malicious' or 'benign'), a supporting explanation ('reason'), and a recommended mitigation ('mitigation').

These results demonstrate that our method enables the model to make accurate security judgments and articulate meaningful defenses—highlighting its potential for fine-grained, explainable threat analysis.

1512
1513
1514
1515
1516
1517
1518
1519
1520
1521
1522
1523
1524
1525
1526
1527
1528
1529
1530
1531
1532
1533
1534
1535
1536
1537
1538
1539
1540
1541
1542
1543
1544
1545
1546
1547
1548
1549
1550
1551
1552
1553
1554
1555
1556
1557
1558
1559
1560
1561
1562
1563
1564
1565

**Output Example 1**

{"name": "hypedrop", "verdict": "Malicious",
"reasoning": "The subgraph for this package contains numerous activities and calls that raise significant red flags and suggest malicious intent:
1. Password Decryption: The use of
'ctypes.windll.crypt32.CryptUnprotectData' indicates an attempt to decrypt sensitive data, which is commonly used to scrape encrypted data like saved passwords.
2. Clipboard Hijacking: The use of 'pyperclip.copy' in combination with 'auto_copy_wallet.address_swap' suggests the possibility of clipboard manipulation, potentially to replace wallet addresses for cryptocurrency theft.
3. Credential Harvesting: Calls like 'os.getlogin', 'winreg.OpenKey', and 'winreg.QueryValueEx' indicate that the package may be gathering system and user information illegitimately.
4. Screen and Cookie Stealing: Nodes like 'PIL.ImageGrab.grab' (screenshot capture) and 'steal_cookies' strongly imply the stealing of private user data.
5. Unauthorized Memory Operations: Calls to
'ctypes.cdll.msvcrt.memcpy' and 'ctypes.windll.kernel32.LocalFree' raise concerns about memory manipulation, which is often associated with malicious payload deployment or data harvesting.
6. Network Activity: The usage of 'urllib.request.Request' and 'urllib.request.urlopen' under the function 'LoadUrlib' may imply exfiltration of data or receiving commands from a remote server.
7. Potential Anti-Debugging: The function 'NoDebugg.sp3cCheq' suggests the package may contain anti-debugging checks to evade detection by security tools or analysts.
8. Execution and Persistence: The presence of 'subprocess.Popen', combined with suspicious function names like 'ZipThings' and system-manipulative calls like 'os._exit', hints at unauthorized script execution or persistence mechanisms.
These behaviors, when taken together, are highly indicative of malicious software designed for data theft, remote control, or system exploitation.",
"mitigation": "The following nodes/calls should be flagged as highest priority for input validation, permission checks, or removal:
1. 'ctypes.windll.crypt32.CryptUnprotectData': Validate any use of sensitive data decryption and ensure it is necessary and authorized.
2. 'pyperclip.copy': Restrict clipboard access to prevent modification of user data.
3. 'PIL.ImageGrab.grab' and 'steal_cookies': Explicitly validate and limit access to screen capture and browser storage to prevent data theft.
4. 'ctypes.cdll.msvcrt.memcpy' and 'ctypes.windll.kernel32.LocalFree': Review all direct memory manipulation for potential abuse.
5. 'urllib.request.Request' and 'urllib.request.urlopen': Monitor outgoing network calls to prevent unauthorized data exfiltration or command-and-control communication.
6. 'os.getlogin', 'winreg.OpenKey', and 'winreg.QueryValueEx': Validate and restrict access to user information and registry keys.
7. 'NoDebugg' Functions: Investigate all anti-debugging mechanisms and ensure they do not attempt to evade legitimate security analyses.
8. 'subprocess.Popen': Confirm that subprocess spawning is securely configured and does not execute unauthorized commands or scripts.
These mitigations are critical to address the highly suspicious nature of the subgraph behavior." }

1566
1567
1568
1569
1570
1571
1572
1573
1574
1575
1576
1577
1578
1579
1580
1581
1582
1583
1584
1585
1586
1587
1588
1589
1590
1591
1592
1593
1594
1595
1596
1597
1598
1599
1600
1601
1602
1603
1604
1605
1606
1607
1608
1609
1610
1611
1612
1613
1614
1615
1616
1617
1618
1619

**Output Example 2**

{"name": "pipcoloradds", "verdict": "Malicious",
"reasoning": "The subgraph structure shows the use of multiple functions and operations that are indicative of malicious behavior. Specifically:
- Data Theft: The presence of 'PIL.ImageGrab.grab' and 'pipcoloradds-1.0.0.pipcoloradds._init_.bc_initial_func.steal_screen' implies potential screen grabbing functionality.
- Token and Credential Stealing: The use of
'ctypes.windll.crypt32.CryptUnprotectData' indicates an attempt to decrypt sensitive data such as stored browser or system credentials. Additionally,
'pipcoloradds-1.0.0.pipcoloradds._init_.bc_initial_func.steal_token' makes use of network calls ('requests.get') and file system traversal ('os.walk'), which may be aimed at stealing access tokens or other sensitive information.
- Clipboard Manipulation: The usage of 'pyperclip.copy' with 'pipcoloradds-1.0.0.pipcoloradds._init_.auto_copy_wallet.address_swap' suggests the possibility of modifying clipboard data, a tactic typically used in cryptocurrency address hijacking attacks.
- Exfiltration of Data: Calls to 'httpx.post'
(pipcoloradds-1.0.0.pipcoloradds._init_.bc_initial_func.ping_on_running) may indicate exfiltration of stolen data to a remote server.
- Code Execution: The presence of 'exec' within 'pipcoloradds-1.0.0.pipcoloradds._init_' raises serious concerns as it can dynamically execute arbitrary code—potentially downloaded or injected during runtime.
- Security Evasion: The
'pipcoloradds-1.0.0.pipcoloradds._init_.NoDebugg' node and behavior suggest an attempt to evade debugging or sandbox analysis by detecting or deterring such environments.
- Cryptography Misuse: The integration of 'fernet.Fernet' may indicate encryption or decryption of stolen data, potentially before exfiltration.
- Persistence Mechanisms: The use of 'os.makedirs' ('pipcoloradds-1.0.0.pipcoloradds._init_.bc_initial_func._init_') and 'shutil.rmtree'
(pipcoloradds-1.0.0.pipcoloradds._init_.bc_initial_func.bc_exit _this) suggests the potential setup or cleanup of malicious files or directories for persistence.
Overall, this combination of functionalities strongly aligns with the behavior of malware, such as information stealers or ransomware.",
"mitigation": "Critical Nodes for Input Validation or Permission Checks:
1. 'ctypes.windll.crypt32.CryptUnprotectData' – This is used to decrypt credentials or sensitive data; ensure it cannot access unauthorized data.
2. 'httpx.post' and 'requests.get' – These are used for exfiltration; validate and restrict network calls.
3. 'PIL.ImageGrab.grab' – This is used for screen capture; restrict screen capture permissions.
4. 'pyperclip.copy' – Prevent unauthorized clipboard manipulation.
5. 'exec' – Deny or heavily sandbox dynamic code execution.
6. 'fernet.Fernet' – Ensure encryption routines are not used maliciously." }

1620
1621

---

### Output Example 3

1622 {"name": "pythonsqliteaddition", "verdict": "Malicious",
1623 "reasoning": "The subgraph structure exhibits multiple indicators of potentially malicious activity:
1624 1. Credential and Data Theft Indicators :
1625 - Calls to 'ctypes.windll.crypt32.CryptUnprotectData' are commonly associated with decrypting stored
1626 credentials (e.g., browser passwords).
1627 - Functions named 'steal_cookies', 'steal_passwords', and 'steal_screen', combined with calls to
1628 'PIL.ImageGrab.grab' (for capturing screenshots) and 're.match' (potentially for parsing sensitive
1629 data), strongly imply an intent to steal user information.
1630 - Usage of 'pyperclip.copy' within 'auto_copy_wallet.address_swap' suggests this action could alter or
1631 steal clipboard data, often done in cryptocurrency wallet hijacking.
1632 2. Use of ctypes for Memory Operations:
1633 - The 'ctypes.cdll.msvcrt.memcpy' function call could manipulate memory directly, adding another
1634 layer of potential obfuscation or tampering.
1635 3. Persistence and Evasion:
1636 - The presence of functions such as 'NoDebugg' and 'NoDebugg.regkey' indicates anti-debugging and
1637 anti-analysis techniques, which are common for malicious software.
1638 - Use of 'os.system' and 'winreg.QueryValueEx' could be employed for modifying the system envi-
1639 ronment or registry keys to establish persistence.
1640 4. System Profiling:
1641 - Multiple calls to 'psutil' functions such as 'psutil.cpu_count', 'psutil.virtual_memory', and 'psu-
1642 til.process_iter' suggest the script profiles the victim's system, which may be used for tailoring attacks
1643 or exfiltration.
1644 5. Potential Ransomware Components:
1645 - The use of 'bc_create_file' and 'ZipThings', as well as 'subprocess.Popen' for zipping or creating
1646 files, combined with the functions above, can be indicative of exfiltration or encryption of user data.
1647 6. Exfiltration and Execution:
1648 - URLs are accessed via 'urllib.request.Request' and 'urllib.request.urlopen', which could be used to
1649 send stolen data or download additional malicious components.
1650 - The 'finished_bc' and 'bc_exit_this' calls ending with 'os._exit' may indicate the script's flow has clear
1651 termination after completing malicious operations.",
1652 "mitigation": "The highest priority nodes or calls for input validation or permission checks are:
1653 1. 'ctypes.windll.crypt32.CryptUnprotectData': Limit access to sensitive decryption functions by re-
1654 stricting permissions.
1655 2. 'PIL.ImageGrab.grab': Disable unauthorized screenshot functionality.
1656 3. Functions related to clipboard modification ('pyperclip.copy'): Restrict clipboard operations.
1657 4. System commands and registry operations ('os.system', 'winreg.QueryValueEx'): Implement strict
1658 permission checks and monitor usage.
1659 5. Network-related calls ('urllib.request.Request', 'urllib.request.urlopen'): Ensure the script isn't
1660 communicating with malicious entities and validate permitted endpoints.
6. 'os._exit': Audit its usage in combination with other suspicious nodes, such as file or data manipu-
lation.
The subgraph strongly indicates malicious intent, and immediate analysis of the full package and its
behavior during runtime in an isolated environment is recommended." }

---

### Output example 4

1662 {"name": "Flask-Bootstrap", "verdict": "Benign",
1663 "reasoning": "The subgraph provided outlines nodes and their interactions within a Flask-Bootstrap
1664 package, which is used to integrate Bootstrap functionality into Flask applications. All the nodes and
1665 edges in the subgraph are consistent with expected behavior in this context. The use of WTForms
1666 renderer and related DOM elements ('div', 'form', 'nav', etc.), alongside utilities like 'flask.url_for',
1667 'markupsafe.Markup', and 'dominate.tags', indicates standard functionalities for rendering templates,
1668 generating URLs, and adding styles or interactivity. There is no evidence of unexpected or suspicious
1669 function calls, network interactions, or operations that could potentially mislead or harm the user.",
1670 "mitigation": "No immediate mitigation is necessary based on this structure. However, as a best prac-
1671 tice:
1672 - Ensure that input validation is implemented on fields rendered by the 'WTFormsRenderer' and pro-
1673 cessed forms to mitigate potential injection attacks.
- Audit 'StaticCDN.get_resource_url' and 'flask.url_for' logic to ensure only safe and validated resource
URLs are generated or included in templates." }

**Output example 5**

{"name": "CausalInference", "verdict": "Benign",
"reasoning": "The subgraph indicates a package primarily focused on causal inference applications, with utilities and tests leveraging statistical methods and random data generation for validation. Most component nodes and edges point towards expected behavior for a package of this type, such as the use of 'numpy.random.multivariate_normal' and 'numpy.random.binomial' for generating random data, and 'distutils.core.setup' for package setup. The connections to various testing nodes and utility functions (e.g., 'test_causal', 'test_matching', etc.) suggest an emphasis on proper validation of internal operations. No obvious malicious indicators like obfuscated code executions, network calls, or actions outside the scope of causal inference computations are present.",
"mitigation": "Although no malicious activity is detected, input validation should be prioritized at the following higher-risk nodes for ensuring package integrity:
1. 'numpy.random.multivariate_normal' and 'numpy.random.binomial': Validate inputs to random data generation functions to ensure parameters are appropriately bounded to prevent system misuse or instability.
2. 'distutils.core.setup': Confirm that the setup script does not introduce unexpected installation behaviors or dependencies.
3. 'causal.sumlessthan', 'calc_att_se', 'log1exp': Ensure robust checks for mathematical operations to prevent improper assertions or computation errors during causal inference calculations." }

