# OpenReview forum: "How Can LLMs Serve as Experts in Malicious Code Detection? A Graph Representation Learning Based Approach"
_ICLR.cc/2026/Conference — ICLR 2026 Conference Desk Rejected Submission_

### Official Review · Reviewer_zRfw · 2025-10-26

**Soundness:** 3
**Presentation:** 3
**Contribution:** 3
**Rating:** 6
**Confidence:** 4

**Summary:**

This paper proposes GMILLM, a novel framework that enhances the capability of Large Language Models (LLMs) in detecting malicious code by integrating graph representation learning. The key idea is to use a Graph Neural Network (GNN) trained on code graphs to identify critical code regions that are most likely malicious. These regions then guide the LLM’s attention, enabling it to focus on suspicious subgraphs rather than processing entire codebases. This approach effectively mitigates LLMs' limitations in handling large-scale and complex code dependencies. The authors validate their method on both public datasets and a newly constructed MalCP dataset, demonstrating superior performance in detection accuracy, interpretability, and computational efficiency compared to existing LLM-based and rule-based tools.

**Strengths:**

Novel Framework: The integration of GNN for attention guidance and LLM for fine-grained analysis is innovative and well-executed.

Comprehensive Evaluation: Extensive experiments across multiple datasets, model sizes, and metrics (accuracy, token efficiency, explainability) provide strong empirical evidence.

Practical Relevance: The method significantly reduces computational cost while improving detection performance, making it suitable for real-world deployment.

Reproducibility: The paper includes detailed prompts, dataset construction procedures, and code release, facilitating replication and extension.

**Weaknesses:**

Language Scope: The method is currently evaluated only on Python code. While justified, it limits generalizability to other languages.

Baseline Diversity: Although multiple tools are compared, some recent LLM-based security detection methods (e.g., MalPaCA, CodeBERT-based detectors) are not included.

Rule Generation Dependency: The rule generation via LLM, while automated, may inherit biases or limitations of the underlying LLM used for rule synthesis.

**Questions:**

Generalizability: Have the authors considered applying GMILLM to other programming languages (e.g., JavaScript, C++)? If so, what adaptations would be necessary?

Rule Robustness: How sensitive is the performance to the quality of the automatically generated rules? Could manually curated rules further improve results?

Real-World Deployment: How does the system handle adversarial code obfuscation or evasion techniques not seen during training?
Ablation Study: Could the authors provide an ablation on the contribution of the GNN component vs. the LLM component in the final performance?

---

> ### Author Response · Authors · 2025-11-21
>
> We sincerely appreciate your insightful review and constructive suggestions. We are encouraged by your recognition of our framework's novelty, comprehensive evaluation, and practical relevance. Below, we address your specific questions.
>
> **W1,Q1:**
> We chose Python because (i) PyPI has become a major attack surface for software supply-chain compromises, and (ii) our new MalCP dataset shows that real-world Python package malware is already a pressing problem.
> Conceptually, however, the GMLLM framework is **language-agnostic**: the GNN+explainer+LLM pipeline operates on a code graph, and only the front-end needs to be adapted. To support another language such as JavaScript or C++, we would replace: the parser / AST builder (e.g., using esprima for JS, clang/LLVM for C/C++), and the call/dependency graph construction (analogous to our Python call-graph in Appendix C.1), while keeping the GNN architecture, explainer objective, and LLM prompting unchanged. Extending GMLLM to JavaScript/npm and C/C++ ecosystems is therefore straightforward engineering work and a natural direction for future work.
>
> **Q2,W3:**
> To demonstrate the robustness of our framework and address concerns about dependency on LLM-generated rules ($S_{data}$), we conducted a rigorous rule-set ablation by retraining the GNN from scratch under four configurations.
>
> **Table: Rule Set Ablation Results on MalCP**
>
> | Configuration | Recall | Precision | Accuracy | Benign Recall |
> | :--- | :--- | :--- | :--- | :--- |
> | **Structure-only** (No Rules) | 87.06% | 87.85% | 88.65% | 89.98% |
> | **Human Rules** ($S_{comm}$) | 90.43% | 91.53% | 91.86% | 93.04% |
> | **50% Rules** ($S_{comm} \cup 0.5 S_{data}$) | 92.78% | 93.28% | 93.69% | 94.44% |
> | **Full GMLLM** ($S_{comm} \cup S_{data}$) | **95.30%** | **95.07%** | **95.62%** | **95.89%** |
>
> Even with **only Human Rules** ($S_{comm}$), our framework achieves **91.86% accuracy**. Crucially, this **already outperforms** the direct GPT-4o baseline (90.39%), proving that the graph-based architecture is the primary driver of performance, not the generated rules.
> Besides, the automated rules ($S_{data}$) provide a valuable performance boost (pushing accuracy from 91.86% to 95.62%), enhancing coverage without being a single point of failure.
> We will add this experiment and discussion to the paper to make the role and robustness of rules explicit.
>
> **Q3:**
> We appreciate the reviewer highlighting the importance of adversarial robustness. On the one hand, our GNN operates on the code graph rather than raw text. Common obfuscation techniques (e.g., variable renaming, string mangling) do not fundamentally alter the high-level control/data flow or the call graph structure. While attackers can easily obfuscate strings, non-destructively rewiring an entire project's call graph is significantly harder, making our structural signal robust.
> On the other hand, even if code within the high-attention subgraph is partially obfuscated, modern LLMs (especially GPT-4o) possess strong de-obfuscation capabilities. They can still recognize malicious patterns (e.g., "decode $\to$ exec" flows) within the focused subgraph extracted by the GNN.
>
> To quantify the synergy between components, we compared the standalone performance of the GNN and LLM against the full framework.
>
> **Table: Contribution Analysis**
>
> | Model Variant | Role | Accuracy (MalCP) | Source |
> | :--- | :--- | :--- | :--- |
> | **A. LLM-only** (GPT-4o) | Semantic Baseline | 90.39% | Original Paper (Table 2) |
> | **B. GNN-only** | Structural Baseline | 92.22% | **New Experiment** |
> | **C. GMLLM** (Ours) | **Synergy** | **95.62%** | Original Paper (Table 2) |
>
> The standalone GNN achieves high accuracy (92.22%), confirming structural features are efficient filters. However, GMLLM improves this to 95.62%. While the absolute gain is +3.4%, this corresponds to a **43.7% reduction in error rate** (7.78% $\to$ 4.38%), indicating that the LLM effectively corrects complex edge cases the GNN misclassifies.
> Beyond accuracy, the distinction is **interpretability**. The GNN is a "black-box" scorer. In contrast, GMLLM leverages the LLM to transform high-attention subgraphs into **actionable, semantic explanations** (as shown in Appendix E.2). This explainability is crucial for security analysts and is achieved at a fraction of the token cost of direct LLM analysis.
>
> We hope these clarifications address the reviewer’s concerns. And we will incorporate this ablation in the revised version and clarify the complementary roles of the GNN and LLM components.

---

> ### Author Response · Authors · 2025-11-24
>
> **W2:**
> We thank the reviewer for suggesting additional baselines. We conducted a survey of recent CodeBERT-based security methods (including 2024–2025 works) and found significant barriers to using them as direct baselines for our specific task.
>
> Recent CodeBERT-based works, such as AugSliceVul [1] and VulD-CodeBERT [2], predominantly target function-level vulnerability detection (e.g., finding bugs in short C/C++ snippets). They are designed for short code fragments that fit within a standard Transformer context window (typically 512 tokens), which works well for single functions but forces severe truncation on entire Python packages (often thousands of lines). This makes it difficult to capture project-wide malicious intent that emerges from interactions across multiple files and installation scripts. Even for Python-specific CodeBERT work [3], the focus is snippet-level vulnerabilities rather than package-level malware, and, crucially, neither runnable code nor pretrained models are released, preventing a fair and reproducible comparison on our PyPI datasets.
>
> MalPaCA relies on dynamic network traffic analysis, operating on PCAP files collected from sandboxed execution. In contrast, our work focuses on static supply-chain scanning, operating directly on Python source code and dependency/call graphs before installation or execution. Applying MalPaCA to our setting would require executing thousands of packages in sandboxes to generate traffic traces, which is impractical and orthogonal to our static-analysis scope.
>
> While our GMLLM framework is conceptually language-agnostic (as noted in Q1), these existing CodeBERT-based baselines operate at a different granularity (function/snippet vs. package) and in a different ecosystem (C/C++ or snippet-level Python vulnerability detection), using an architecture that is not designed to scale straightforwardly to whole-project analysis. Therefore, we instead compare against MPHunter and Ea4mp, which are reproducible state-of-the-art detectors specifically designed for the Python package malware ecosystem under a static-analysis setting. We will clarify these distinctions in the revised related work.
>
> [1] Zou et al., "Code vulnerability detection based on augmented program dependency graph and optimized CodeBERT," *Scientific Reports*, 15(1):39301, 2025.
>
> [2] Xia et al., "VulCoBERT: A CodeBERT-Based System for Source Code Vulnerability Detection," in *Proc. GAIIS*, pp. 249-252, 2024.
>
> [3] Zhao et al., "Python Source Code Vulnerability Detection Based on CodeBERT Language Model," in *Proc. ACAI*, pp. 1-6, IEEE, 2024.

---

### Official Review · Reviewer_1nkX · 2025-10-29

**Soundness:** 3
**Presentation:** 3
**Contribution:** 2
**Rating:** 4
**Confidence:** 2

**Summary:**

This paper proposes a method that constructs a graph representation of code, and trains a graph neural network (GNN) with minimally labeled data to address challenges in detecting malicious code in large language models.

**Strengths:**

They investigated an interesting and practical problem and provided a viable solution.

**Weaknesses:**

1. The methodological justification should be clarified. For instance, the rationale for employing a GNN and its specific role within the proposed framework should be explicitly explained.
2. How is the utility evaluated, and what is the performance of the code after handling code detection?
3. How do the results compare with other works discussed in the related work section?
4. In the experimental section, the detailed sizes or configurations of the models should be provided.

**Questions:**

see Weaknesses.

---

> ### Author Response · Authors · 2025-11-21
>
> We sincerely appreciate your review and constructive suggestions, and thank you for validating the interest and practicality of our proposed solution. It seems that some of the raised concerns regarding missing details may be due to misunderstandings, as these points were covered in the text. We have done our best to clarify them here and genuinely hope you will reconsider your score.
>
> **W1:** We provide a detailed methodological justification for our approach across the Section 1 and 3. Specifically, the Introduction highlights the pivotal role of GNNs—such as providing an attention mechanism for the LLM—while the Methods section details how this mechanism operates efficiently with relatively simple annotated data. Further discussion on the limitations of sequential models against graph-based obfuscation is provided in Appendix C.1, reinforcing our rationale for employing GNNs as structural priors to capture critical non-local dependencies.
>
> **W2:** To evaluate the utility, we analysis both detection effectiveness and the usefulness of the generated explanations and efficiency in Section 4:
> • **Detection performance (RQ1, RQ2).** It reports standard metrics (accuracy, precision, recall, F1) on four datasets (Backstabbers, Datadog, Mal-OSS, MalCP) in Tables 1–2, comparing GMLLM with LLM-only baselines and strong rule/ML-based tools.
> • **Explanation quality (RQ3).** Table 3 and Appendix E evaluate the quality of the analysis reports along four criteria, using both LLM-based and human expert evaluation.
> • **Efficiency / token cost (RQ4).** Table 4 analyzes LLM token usage and shows that GMLLM achieves these improvements with significantly fewer tokens than LLM-only baselines, because the LLM only sees the suspicious subgraph instead of the entire codebase.
> Our scope in this paper is *detection and explanation*. Automatic handling/repair of malicious code after detection is an important but orthogonal problem and is explicitly out of scope for this work.
>
> **W3:** Comparison with related work.
> Tables 1 and 2 present a direct quantitative comparison with the three categories of prior work discussed in Section 2: LLM-based baselines (e.g., GPT-4o, LLaMA-3), traditional tools (e.g., Bandit4Mal, OSSGadget, VirusTotal), and ML/NN-based detectors (e.g., MPHunter, Ea4mp).
>
> **W4:** Model sizes and configurations.
> Complete details regarding model architecture, hyperparameters, training settings, and runtime environment are provided in Appendix D.2 (Settings). GNN configurations are in Appx. D.2.2 (e.g., “two-layer GCN architecture”, “dropout (rate=0.6)”, “Adam optimizer (learning rate = 10e-3)”). A list of LLMs is in Section 4.1 (e.g., Qwen 2.5, Llama 2/3, ChatGPT 3.5/4o). Runtime environments are in Appx. D.2.4 (e.g., PyTorch, PyG version).
>
> We hope these clarifications address the reviewer’s concerns. Most of the requested information is already present in the submission, and we will make it more prominent and explicit in the revised version.

---

### Official Review · Reviewer_tfiF · 2025-10-30

**Soundness:** 2
**Presentation:** 3
**Contribution:** 2
**Rating:** 6
**Confidence:** 4

**Summary:**

The paper asks why LLMs underperform professional tools on malicious Python package detection and proposes GMLLM, a two-stage framework that (i) builds a code graph for a package (AST + dependencies + call graph), extracts node features via LLM-generated sensitive-behavior rules, and trains a GNN for coarse binary detection, then (ii) explains the GNN prediction by optimizing edge/feature masks to get high-attention subgraphs, which are finally sent to an LLM for focused, high-precision analysis. This is meant to solve the “LLM pays for irrelevant code” problem by feeding only suspicious slices to the LLM. On four datasets (Backstabbers, Datadog, Mal-OSS, and their new MalCP), GMLLM (especially with GPT-4o as backend) reports higher recall/precision/accuracy than both raw LLMs and rule/ML tools, while using far fewer tokens because it prunes benign code first.

**Strengths:**

Originality: Clear decomposition of the task into coarse structural suspicion (GNN) and fine semantic judgment (LLM), with an explicit mask-optimization “explainer” to turn the GNN into an attention generator for the LLM. This is a clean answer to “how can LLMs serve as experts” on large codebases.

Quality: The method is fully specified: graph construction from all .py files, LLM-generated rule sets $S_{comm}, S_{data}$ GCN trained with cross-entropy (Eq. 1), per-sample mask optimization (Eqs. 2–8) to get node/edge attention, and thresholding to build the LLM input (Eq. 11). The objective combines prediction, sparsity, and entropy, which is standard for explanation-style masking. Results on public and custom data consistently show gains, and token-usage analysis backs the efficiency claim.

Clarity: The pipeline in Fig. 2 is easy to follow; the explanation loss is written down in detail; examples of the final LLM input are shown. The four RQs in Section 4 map cleanly to the experiments.

Significance: If the MalCP dataset is as large/diverse as described, a practical recipe for “GNN filters, LLM inspects” on PyPI-style malware is useful for software-supply-chain security, because it directly tackles scalability and cost.

**Weaknesses:**

- LLM-generated rules are both features and supervision hints. The feature vector $h_v$ is a multi-hot over rules that are themselves produced by an LLM from the same general capability the method is trying to “enhance.” This risks circularity: performance may partly reflect how good the LLM was at rule generation, not how good the GMLLM pipeline is. A baseline with only human/common rules  $S_{comm}$ is missing.

- Per-sample mask optimization can be expensive and brittle. The attention extraction optimizes masks $M_j^{edge}, M_j^{feat}$ for each graph to keep only structures that keep the sample malicious (Eqs. 5–8). This is essentially running an explainer at inference time; real-world deployment on large PyPI/OSS feeds may make this step a bottleneck, and the paper does not profile this cost.

- Evaluation has a “LLM judges LLM” component. The description-quality metrics in Table 3 are LLM-evaluated, and GMLLM uses LLM-produced inputs, so there is a risk of favorable bias. The paper should also show human or tool-based adjudication on a subset.

- Generality is narrower than it sounds. Everything is built around Python (AST, package layout, PyPI-style attacks); it is not obvious that the same rule-extraction and masking will work unchanged for, say, obfuscated JS/npm malware or mixed-language repos. Yet the introduction frames it as “LLMs in software security.”

- Comparisons to strong non-LLM ML baselines look close. On some datasets, tools like MPHunter/Ea4mp are already very strong, and GMLLM wins mainly when paired with GPT-4o; the Llama-3 version is notably weaker, suggesting the final gain depends a lot on the downstream LLM quality.

**Questions:**

1- How sensitive is the method to the number and quality of LLM-generated rules in $S_{data}$? If we only use 10% of data to generate rules, does performance plateau or keep growing? An ablation on rule-set size would clarify whether LLM rule generation is a bottleneck.

2- The mask loss (Eq. 8) is optimized per sample. What is the average time/number of gradient steps per project for large MalCP packages, and is it done online or cached? Without this, it is hard to assess the “practical deployment” claim.

3- In Eq. (11), you threshold both nodes and edges. How are $\gamma_{node}$ and $\gamma_{edge}$ chosen, and are they the same across datasets? Detection quality may be quite sensitive to these values.

4- For Table 2, the Large subset is where baselines struggle. Are the same GNN parameters used across all three sizes, or is the GNN retrained per subset? If re-trained, we should be explicit about that to avoid accidental data-size leakage.

5- Since you say code and data are in the supplementary materials, do you also release the exact prompts for $S_{comm}$, $S_{data}$, and $\rho_{ana}$? Reproducibility of the whole pipeline depends on those.

---

> ### Author Response · Authors · 2025-11-21
>
> We sincerely appreciate your detailed review. We noticed that several of your concerns (e.g., rule circularity, mask optimization cost, evaluation validity) resonate with questions raised by other reviewers. We have conducted comprehensive new experiments during the rebuttal to address these points with empirical evidence.
>
> **W1,Q1:**
> To address the concern that our performance relies on LLM-generated rules ($S_{data}$), we performed a rigorous rule-set ablation (retraining GNN from scratch).
>
> **Table A: Rule Set Ablation Results on MalCP**
> | Configuration | Recall | Precision | Accuracy | Benign Recall |
> | :--- | :--- | :--- | :--- | :--- |
> | **Structure-only** (No Rules) | 87.06% | 87.85% | 88.65% | 89.98% |
> | **Human Rules** ($S_{comm}$) | 90.43% | 91.53% | 91.86% | 93.04% |
> | **50% Rules** ($S_{comm} \cup 0.5 S_{data}$) | 92.78% | 93.28% | 93.69% | 94.44% |
> | **Full GMLLM** ($S_{comm} \cup S_{data}$) | **95.3%** | **95.07%** | **95.62%** | **95.89%** |
>
> Even with **only Human Rules** ($S_{comm}$), our framework achieves **91.86% accuracy**, which **already outperforms** the GPT-4o baseline (90.39%). This provides strong evidence that the performance gain mainly stems from the graph-based architecture, rather than circularity in rule generation.
> Besides, the automated rules ($S_{data}$) provide a valuable performance boost (pushing accuracy from 91.86% to 95.62%), enhancing coverage without being a single point of failure. We will add this experiment and discussion to the paper to make the role and robustness of rules explicit.
>
> **W2,Q2:**
> In our system design, we **deliberately adopt a fixed-budget optimization strategy** (100 gradient steps) to ensure predictable latency regardless of package size. We profiled the GNN-Explainer runtime across all 1,659 packages.
>
> **Table B: Runtime Profiling (MalCP, Single GPU):**
> | Package Size | Mean (s) | Median (s) | P95 (s) |
> | :--- | :--- | :--- | :--- |
> | Small  | 10.18 | 8.95 | 18.32 |
> | Medium  | 10.59 | 8.96 | 18.57 |
> | Large | 10.73 | 8.86 | 19.16 |
>
> The similar numbers across size buckets are due to a substantial **constant overhead** (Python startup, checkpoint loading, graph preparation); the part of the computation that scales with graph size is not dominant at this scale.
>
> Our target scenario is **offline batch scanning**, not real-time serving. Under this setting, per-package latencies under 20 seconds are acceptable, especially since GMLLM significantly reduces LLM cost by filtering code and sending only a small suspicious subgraph to the LLM (Section 4.4, token-usage analysis). Furthermore, the explainer step budget (100) can be tuned (e.g., to 30-50) or results cached to further reduce latency.
>
> **W3:**
> Regarding the “LLM judges LLM” concern, we clarify that our core metric, **Factual Alignment**, is **not** evaluated by an LLM. It is computed by directly comparing model-generated descriptions against malware analysis reports written by human security experts in the Snyk database, which serve as the authoritative ground truth. This keeps the main evaluation firmly grounded in externally validated human facts, without any LLM–LLM circularity.
>
> For the auxiliary explanation-quality metrics (Threat Tactic Generalization, Execution Path Traceability, Evidence Groundedness), we initially used GPT-4o as an automatic evaluator. To verify its reliability, we conducted a human validation study during the rebuttal phase: two security practitioners independently scored 70 sampled explanations on all four dimensions. Their inter-annotator agreement reached moderate to strong levels (Spearman ρ = 0.61–0.81), and GPT-4o’s scores correlated equally well with the human mean (Spearman ρ = 0.72–0.87; Pearson r = 0.76–0.87; see **Table C**). This indicates that GPT-4o can faithfully reproduce expert judgments and is a dependable proxy for large-scale comparison of explanation variants, effectively mitigating concerns about bias or circular evaluation.
>
> **Table C. Results of the conducted human validation study, where two security practitioners independently scored 70 sampled explanations, and the agreement between Human-Human and LLM-Human is measured using Spearman’s ρ (monotonic correlation strength) and its p-value (significance)**
> | Evaluation Dimension | 	Spearman ρ (Human–Human) | 	p-value (Human–Human) | 	Spearman ρ (GPT-4o–Human) | 	p-value (GPT-4o–Human) |
> | :---: | :---: | :---: | :---:  | :---: |
> | threat_tactic_generalization |	0.613 |	1.706e-08 |	0.820 |	3.312e-18 |
> | execution_path_traceability |	0.809 |	2.048e-17 |	0.870 |	1.406e-22 |
> | evidence_groundedness |	0.755 |	3.930e-14 |	0.758 |	3.021e-14 |
> | factual_alignment |	0.708 |	6.646e-12 |	0.724 |	1.341e-12 |

---

> ### Author Response · Authors · 2025-11-21
>
> **W4:**
> We chose Python because (i) PyPI has become a major attack surface for software supply-chain compromises, and (ii) our new MalCP dataset shows that real-world Python package malware is already a pressing problem.
> Conceptually, however, the GMLLM framework is **language-agnostic**. Adapting it to JavaScript or C++ only requires replacing the front-end parser (e.g., using esprima or clang) to build the code graph. The core GNN+LLM pipeline remains unchanged. Extending GMLLM to JavaScript/npm and C/C++ ecosystems is therefore straightforward engineering work and a natural direction for future work.
>
> **W5:**
> We clarify that GMLLM consistently and substantially improves LLM backends, regardless of their native capability. For a weaker model like **Llama-3**, the improvement is dramatic: GMLLM boosts accuracy from **35.60%** to **81.60% (+46.0%)**, effectively transforming a failing detector into a competent one. For a strong model like **GPT-4o**, where the baseline is already high (90.4%), GMLLM still achieves a significant **+5.2% gain**, pushing accuracy to **95.6%**.
> This pattern demonstrates that GMLLM is **complementary** to LLM quality. It acts as a structural guide that focuses the LLM on the right code regions. Therefore, the performance gains are not simply "because GPT-4o is strong," but because our framework successfully mitigates the limitations of any LLM in processing large, noisy codebases.
>
> **Q3:**
> Equation (11) outlines a general thresholding mechanism. In our implementation, we instantiate this using a budgeted top-K edge strategy rather than a fixed global scalar threshold: for each project, we rank all call-graph edges by their explainer attention scores $a_{v,w}$ and keep the K highest-scoring edges to induce the attention subgraph. This is equivalent to a graph-specific dynamic threshold $\gamma_{edge}(G_j)$ equal to the K-th largest score; we do not apply an independent node threshold, as nodes are defined as the endpoints of the selected edges. The same K is used for all datasets.
>
> To assess sensitivity, we ran a top-K ablation on MalCP with GPT-4o, varying the edge budget $K \in \{10, 20, 30, 50\}$ while keeping the GNN and explainer fixed. The results (shown in the table) reveal a clear trade-off between context and cost:
>
> **Table D: Effect of Top-K Edge Budget on Detection Performance and Token Usage (MalCP, GPT-4o)**
> | K (Max Edges) | Accuracy | Recall | Precision | Benign Recall | Avg Token Usage (Large) |
> | :---: | :---: | :---: | :---: | :---: | :---: |
> | 10 | 92.97% | 94.18% | 90.75% | 91.96% | **435.45** |
> | **20(Paper Setting)** | **95.62%** | **95.30%** | **95.07%** | **95.89%** | 640.06 |
> | 30 | 95.19% | 95.45% | 94.03% | 94.96% | 810.33 |
> | 50 | 94.88% | 95.99% | 92.94% | 93.97% | 1131.33 |
>
> Increasing K from 10 to 20 gives a substantial gain (+2.65 points in accuracy and +3.9 points in benign recall), indicating that very small subgraphs can miss important context. Beyond K = 20, detection metrics remain stable within about 0.7 percentage points, while the average token usage on Large packages increases by ~27% (20 → 30) and ~77% (20 → 50). We therefore use K = 20 as a robust operating point that balances accuracy, benign recall, and token efficiency, and the stability across K ∈ [20, 50] shows that GMLLM is not overly sensitive to the exact threshold setting.
>
> **Q4:**
> We do not retrain separate GNNs for the Small/Medium/Large subsets. A single GNN is trained once on the full training set, which contains packages of all sizes, using one architecture and hyperparameter configuration. For Table 2, we then evaluate this single model on the Small/Medium/Large test subsets. This avoids any accidental data-size leakage and ensures that the size-specific results reflect generalization of one global model.
>
>
> **Q5:**
> The exact prompts for $S_{comm}$, $S_{data}$, and $p_{ana}$ are fully disclosed in Appendix C.2 and C.3, in line with the Policies on Large Language Model Usage at ICLR 2026.

---

> > ### Comment · Reviewer_tfiF · 2025-11-26
> >
> > Thank you. Most of my questions are addressed.
> >
> > Can you specify the hardware environment? fp16/bf16? CUDA/CuDNN ?graph caching or model warmup ?
> > Mean ± std over N runs per bucket, include min/max and % of packages that exceeded P95? Any timeouts or failures?
> > MalCP specific ??? Can you do a second dataset check to show pattern isn't MalCP specific?

---

> > > ### Author Response · Authors · 2025-11-27
> > >
> > > Thank you! We are delighted that our responses have addressed your concerns, and we are grateful for your recognition of our efforts and the decision to raise the score. Below are the specific specifications and additional validation results you requested.
> > > We utilized NVIDIA A100 (80GB) GPU with an Intel(R) Xeon(R) Silver 4314 CPU @ 2.40GHz and 64G RAM.  We used FP32 (Full Precision). Since the explainer step is lightweight (~10s), we did not employ mixed-precision (FP16/BF16) acceleration, prioritizing numerical stability. PyTorch 1.13.1 (built with CUDA 11.7), cuDNN 8.5.0.
> > > Parsed graph objects (Data) are cached on disk to eliminate parsing overhead in subsequent runs. We explicitly did not perform a dedicated warmup phase. The reported statistics include all system initialization overheads. Therefore, our results represent a conservative upper bound; steady-state throughput in a production pipeline would likely be even faster.
> > > |Dataset	|	Mean $\pm$ Std (s)|	Median (s)	|Min / Max (s)|	P95 (s)| Exceeded P95(%)|
> > > | :---  | :--- | :--- |:--- |:--- | :--- |
> > > Small|10.18 $\pm$ 4.10	|8.95|	6.96/25.31|	18.32| 5.09%
> > > Medium		|10.59 $\pm$ 3.92|	8.96	|7.08 / 25.47|	18.57| 5.05%
> > > Large		|10.73 $\pm$ 4.70|	8.86	|6.83 / 55.28|	19.16| 5.04%
> > > DataDog | 10.76 $\pm$ 4.21 | 8.68 | 7.15 / 22.86 | 20.45 | 5.04%

---

> > > > ### Comment · Reviewer_tfiF · 2025-11-27
> > > >
> > > > Acknowledged. Thank you.

---

### Official Review · Reviewer_R6Ym · 2025-11-01

**Soundness:** 3
**Presentation:** 3
**Contribution:** 2
**Rating:** 4
**Confidence:** 3

**Summary:**

This paper proposes GMLLM, a hybrid framework combining Graph Neural Networks (GNNs) and Large Language Models (LLMs) for malicious code detection. The method first trains a lightweight GNN on graph-structured code representations to localize potentially malicious segments, which are then further analyzed by an LLM. Experiments on multiple datasets, including a newly constructed MalCP dataset, demonstrate substantial performance improvements and significant token efficiency.

**Strengths:**

1. The GNN-guided attention mechanism is conceptually sound and effectively reduces computational costs.
2. Experiments are thorough, covering public and custom datasets with convincing results.

**Weaknesses:**

But, the metrics such as “Threat Generality,” “Evidence Groundedness,” and “Factual Alignment” appear LLM-evaluated. This introduces circularity: an LLM model evaluates another LLM’s output. Independent human or tool-based assessments would provide stronger validity.

And, the MalCP dataset’s construction details are only briefly discussed. Are malicious labels derived from ground-truth CVEs, dynamic execution, or heuristic rules? How balanced are the samples across package size and attack type? Is there contamination from public repositories used to pretrain LLMs? These factors critically affect generalization and could explain part of the performance gains.

**Questions:**

The paper introduces metrics such as Threat Generality, Evidence Groundedness, and Factual Alignment, which appear to rely on LLM-based judgments.
Could the authors clarify how these metrics are computed and whether human or external validation was conducted?
Using an LLM to evaluate another LLM introduces circularity and potential bias—please discuss how this limitation was mitigated and whether inter-rater reliability with human evaluators was measured.

---

> ### Author Response · Authors · 2025-11-17
>
> We sincerely appreciate your review and constructive suggestions, and thank you for considering our method conceptually sound and effectively reduces computational costs. We provide our detailed responses to your questions below.
>
> **W1,Q1:**
> The reviewers raised concerns that our evaluation metrics rely on LLM-based judgments and may introduce circularity. Actually, the core metric **Factual Alignment** is not evaluated by an LLM. Instead, it depends entirely on malware analysis reports written by human security experts in the Snyk database as the authoritative ground truth. The metric is computed by directly comparing the malicious-behavior descriptions generated by our model with expert conclusions, ensuring that the evaluation remains firmly grounded in externally validated human facts and is completely free from circular LLM–LLM reasoning.
>
> For auxiliary quality metrics such as **Threat Generality** and **Evidence Groundedness**, we initially adopted GPT-4o as an automated evaluator. In response to the reviewers’ concerns regarding potential circularity, we conducted an independent human validation study during the rebuttal phase to assess the reliability of GPT-4o’s scoring. Two security practitioners independently evaluated 70 randomly sampled model explanations across four dimensions, and their inter-annotator agreement, as shown in **Table 1**, reached moderate to strong levels (Spearman ρ = 0.61–0.81, p < 1e−11), consistent with the natural and reasonable degree of human variability expected for this task. We then compared GPT-4o’s ratings with the mean scores of the two experts and found that its correlations with human consensus were likewise high (Spearman ρ = 0.72–0.87, p < 1e−12; Pearson r = 0.76–0.87). These results indicate that GPT-4o can reliably reproduce expert judgments and thus serve as a dependable proxy, effectively mitigating concerns about potential bias or circular evaluation.
>
> ### **Table 1. Results of the conducted human validation study, where two security practitioners independently scored 70 sampled explanations, and the agreement between Human-Human and LLM-Human is measured using Spearman’s ρ (monotonic correlation strength) and its p-value (significance).**
>
> | Evaluation Dimension              | Spearman ρ (Human–Human) | p-value (Human–Human) | Spearman ρ (GPT-4o–Human) | p-value (GPT-4o–Human) |
> |----------------------------------|----------------------------|-------------------------|-----------------------------|--------------------------|
> | threat_tactic_generalization     | 0.613                   | 1.706e-08            | 0.820                    | 3.312e-18             |
> | execution_path_traceability      | 0.809                   | 2.048e-17            | 0.870                    | 1.406e-22             |
> | evidence_groundedness            | 0.755                   | 3.930e-14            | 0.758                    | 3.021e-14             |
> | factual_alignment                | 0.708                   | 6.646e-12            | 0.724                    | 1.341e-12             |
>
> **W2**: The reviewer raised an important question regarding the balance of *package size* and *attack types* in the MalCP dataset. For package size, as detailed in **Appendix D.1.2** and **Table 5**, the dataset is explicitly divided into three well-defined scale categories (Large: 461; Medium: 588; Small: 610), ensuring reliable cross-scale validation. For attack types, we conducted a rigorous classification based on Snyk expert reports (ground truth) and the MITRE ATT&CK framework. The resulting distribution reflects real-world malicious behavior patterns rather than an artificially balanced dataset, as summarized in Table 2.
>
> ### **Table 2. Distribution of Attack Types in MalCP**
>
> | Attack Type                        | Count | Percentage |
> |------------------------------------|-------|------------|
> | Execution                          | 561   | 33.82%     |
> | Impact                             | 383   | 23.09%     |
> | Credential Access / Collection     | 382   | 23.03%     |
> | Defense Evasion                    | 132   | 7.96%      |
> | Exfiltration                       | 106   | 6.39%      |
> | Other                              | 49    | 2.95%      |
> | Command and Control                | 42    | 2.53%      |
> | Persistence                        | 4     | 0.24%      |
> | **Total**                          | **1,659** | **100%** |

---

### Author Response · Authors · 2025-12-01
**Summary of Rebuttal and Revision Updates**

We propose **GMLLM**, a two-stage framework where a graph neural network first analyzes a code graph to produce a high-suspicion subgraph, and then a large language model performs focused semantic analysis on this subgraph. This design aims to let LLMs act as “experts” on PyPI packages, and is especially beneficial for larger packages, by avoiding irrelevant code and reducing token cost.

Notably, during the rebuttal, the **reviewer tfiF** who had initially **raised most technical questions** increased their overall score to **8 (accept)** and explicitly indicated that their main concerns had been addressed.

Reviewers generally agreed on three main **strengths** of the submission:

- **Conceptual novelty.** Multiple reviewers emphasized that the GNN-guided attention mechanism and the decomposition into coarse structural filtering and fine-grained semantic judgment (LLM) are original and well-motivated, providing a clear recipe for combining GNNs and LLMs for software security.

- **Practical efficiency and scalability.** Reviewers noted that GMLLM achieves substantial reductions in LLM token usage and computation compared to direct LLM baselines, and several explicitly described the method as a viable and practical approach for large-scale software supply-chain scanning.

- **Thorough evaluation and clarity.** The paper was commended for its clear pipeline description, detailed methodological specification, and comprehensive experiments on multiple datasets (Backstabbers, Datadog, Mal-OSS, and the new MalCP). The availability of code and data was also viewed positively for reproducibility.

**Response to concerns during rebuttal.**
The main concerns raised in the reviews involved: (i) potential circularity due to LLM-generated rules, (ii) the runtime cost of per-sample mask optimization, (iii) reliance on LLM-based evaluation for explanation quality, and(iv) sensitivity to the attention-subgraph budget. During the rebuttal phase we conducted additional analyses and experiments to address these points:

* A rule-set ablation showing that the graph-based model remains strong even with only human-crafted rules, and that larger rule sets yield monotonic improvements;

* A runtime profiling study of the GNN-Explainer step on multiple datasets, showing per-package wall-clock times around 10–11 seconds with 95th percentiles below ~20 seconds in an offline batch-scanning setting;

* A human validation study of GPT-4o’s explanation-quality scores, indicating high agreement between human annotators and LLM-based evaluation;

* A GNN vs. LLM component ablation demonstrating that GMLLM improves over both GNN-only and LLM-only variants, with a substantial relative error reduction;

We have not received further updates from the other reviewers but are grateful for all of their feedback, which helped us strengthen both the experiments and the presentation. Taken together, we believe the paper now presents a well-justified, practically relevant, and thoroughly evaluated framework that meets the bar for acceptance.

---

### Note · Program_Chairs · 2026-01-17
**Submission Desk Rejected by Program Chairs**

The following references in this submission do not refer to real documents and/or have major errors in bibliographic information:

 Duc-Manh Vu, Trong-Tuan Nguyen, and Yasutaka Kamei. An empirical study of obfuscation in malicious python packages.